# Breakage of the oligomeric CaMKII hub by the regulatory segment of the kinase

Deepti Karandur[1,2,3†], Moitrayee Bhattacharyya[1,2,3†‡], Zijie Xia[4†§], Young Kwang Lee[1,2,4#], Serena Muratcioglu[1,2,3], Darren McAffee[1,2,4], Ethan D McSpadden[1,2,3¶], Baiyu Qiu[1,2,3**], Jay T Groves[1,2,4,5], Evan R Williams[2,4*], John Kuriyan[1,2,3,4,5]*

[1]Department of Molecular and Cell Biology, University of California, Berkeley, Berkeley, United States; [2]California Institute for Quantitative Biosciences (QB3), University of California, Berkeley, Berkeley, United States; [3]Howard Hughes Medical Institute, University of California, Berkeley, Berkeley, United States; [4]Department of Chemistry, University of California, Berkeley, Berkeley, United States; [5]Physical Biosciences Division, Lawrence Berkeley National Laboratory, Berkeley, United States

**\*For correspondence:**
erw@berkeley.edu (ERW);
jkuriyan@mac.com (JK)

[†]These authors contributed equally to this work

**Present address:** [‡]Department of Pharmocology, Yale University, New Haven, United States; [§]Claros Technologies Inc, St. Paul, United States; [#]Department of Chemistry and Biochemistry, San Diego State University, San Diego, United States; [¶]IDEAYA Biosciences, South San Francisco, United States; [**]Columbia University, New York, United States

**Abstract** Ca$^{2+}$/calmodulin-dependent protein kinase II (CaMKII) is an oligomeric enzyme with crucial roles in neuronal signaling and cardiac function. Previously, we showed that activation of CaMKII triggers the exchange of subunits between holoenzymes, potentially increasing the spread of the active state (Stratton et al., 2014; Bhattacharyya et al., 2016). Using mass spectrometry, we show now that unphosphorylated and phosphorylated peptides derived from the CaMKII-$\alpha$ regulatory segment bind to the CaMKII-$\alpha$ hub and break it into smaller oligomers. Molecular dynamics simulations show that the regulatory segments dock spontaneously at the interface between hub subunits, trapping large fluctuations in hub structure. Single-molecule fluorescence intensity analysis of CaMKII-$\alpha$ expressed in mammalian cells shows that activation of CaMKII-$\alpha$ results in the destabilization of the holoenzyme. Our results suggest that release of the regulatory segment by activation and phosphorylation allows it to destabilize the hub, producing smaller assemblies that might reassemble to form new holoenzymes.

## Introduction

Ca$^{2+}$/calmodulin-dependent protein kinase II (CaMKII) is an oligomeric serine/threonine kinase that is important in neuronal signaling and cardiac function (*Lisman et al., 2002*; *Kennedy, 2016*; *Bhattacharyya et al., 2020a*). Each subunit of CaMKII has an N-terminal kinase domain that is followed by a regulatory segment and an unstructured linker that leads into a C-terminal hub domain (*Figure 1A,B*). The regulatory segment blocks the substrate-binding site of the kinase in the autoinhibited state of CaMKII (*Rosenberg et al., 2005*; *Rellos et al., 2010*), and this inhibition is released by the binding of Ca$^{2+}$/calmodulin (Ca$^{2+}$/CaM) to a calmodulin-binding element within the regulatory segment (*Figure 1B*; *Ikura et al., 1992*; *Meador et al., 1992*; *Meador et al., 1993*; *Rellos et al., 2010*). The hub domains of CaMKII associate to form a ring-shaped scaffold containing twelve or fourteen subunits, around which the kinase domains are arranged (*Figure 1A*; *Hoelz et al., 2003*; *Rellos et al., 2010*; *Chao et al., 2011*; *Myers et al., 2017*). Mutations in CaMKII affect learning and memory formation in mice (*Silva et al., 1992*; *Giese et al., 1998*; *Elgersma et al., 2002*), and mutations in human CaMKII have been identified in patients with cognitive and developmental impediments (*Robison, 2014*; *Küry et al., 2017*; *Chia et al., 2018*).

The activation of CaMKII by Ca$^{2+}$/calmodulin triggers the co-localization of subunits from different CaMKII holoenzymes, indicating that activation results in subunits being exchanged between

holoenzymes (*Stratton et al., 2014*; *Bhattacharyya et al., 2016*). This conclusion was based on the results of experiments in which two samples of CaMKII were labeled separately with fluorophores of two different colors, mixed, and then activated. Single-molecule visualization showed activation-dependent co-localization of the two fluorophores (*Stratton et al., 2014*). In other experiments, separate CaMKII samples were labeled with FRET donor and acceptor fluorophore pairs, respectively, and mixing these samples after activation led to increased FRET compared to mixing unactivated samples (*Bhattacharyya et al., 2016*). These data are also consistent with subunit exchange. Colocalization of differently labeled subunits was also observed when unactivated CaMKII holoenzymes were mixed with activated ones. Importantly, these experiments showed that activated CaMKII holoenzymes could phosphorylate subunits of unactivated ones, thereby spreading the activation signal (*Stratton et al., 2014*).

The regulatory segment of CaMKII has been shown to be important for subunit exchange (*Stratton et al., 2014*; *Bhattacharyya et al., 2016*). After activation, autophosphorylation of Thr 286 in the regulatory segment prevents it from re-binding to the kinase domain, thereby conferring $Ca^{2+}$/CaM independence (autonomy) on CaMKII (*Miller et al., 1988*; *Thiel et al., 1988*; *Lou and Schulman, 1989*). Autophosphorylation of two other sites, Thr 305 and Thr 306, prevents the binding of $Ca^{2+}$/CaM to the regulatory segment (*Colbran, 1993*). Thus, we expect that activation is accompanied by the release of at least some fraction of the regulatory segment from the kinase and from $Ca^{2+}$/CaM.

A constitutively active variant of CaMKII-α (T286D) undergoes spontaneous subunit exchange without activation by $Ca^{2+}$/CaM (*Stratton et al., 2014*). Mutation of the calmodulin-binding element of the regulatory segment in this variant eliminated subunit exchange, which suggests that the calmodulin-binding element might interact with the hub and destabilize it, thereby promoting subunit exchange (*Stratton et al., 2014*). Consistent with this, a construct consisting of just the hub and the regulatory segment was shown to undergo spontaneous subunit exchange (*Bhattacharyya et al., 2016*). Recent differential scanning calorimetry and mass photometry experiments have shown that while the isolated hub is very stable, the holoenzyme, especially with the linker present, is unstable and undergoes dissociation (*Torres-Ocampo et al., 2020*).

A prominent feature of the hub assembly is the presence of grooves located at the inter-subunit interfaces. These grooves are lined by negatively-charged residues, and we proposed that positively-charged residues in the calmodulin-binding element might recognize these negatively-charged residues (*Bhattacharyya et al., 2016*). Each interfacial groove between subunits contains the uncapped edge of a β-sheet, and peptide segments have been shown to dock within the groove by forming an additional strand of this β-sheet (*Chao et al., 2011*; *Bhattacharyya et al., 2016*). A peptide with the sequence of the calmodulin-binding element has been shown to bind weakly to the hub, with a dissociation constant of ~90 μM, but destabilization of the hub was not studied in those experiments (*Bhattacharyya et al., 2016*).

In this paper, we present the results of investigations into interactions between the regulatory segment and the hub, and the effect of activation on the stability of the holoenzyme. We used electrospray ionization mass spectrometry to show that peptides derived from the regulatory segment can bind to and destabilize the dodecameric or tetradecameric CaMKII-α hub assembly, releasing smaller oligomers of the hub domain. We generated a 13 μs molecular dynamics trajectory of the dodecameric hub of CaMKII-α, which shows that two regulatory segments bind spontaneously to the hub at two inter-subunit interfaces. This docking is facilitated by large distortions in the structure of the hub, which reflect an intrinsic feature of hub dynamics, as shown by a normal mode calculation. Finally, we use single-molecule TIRF microscopy to show that activation of CaMKII-α expressed in mammalian cells is accompanied by apparent destabilization of the holoenzyme.

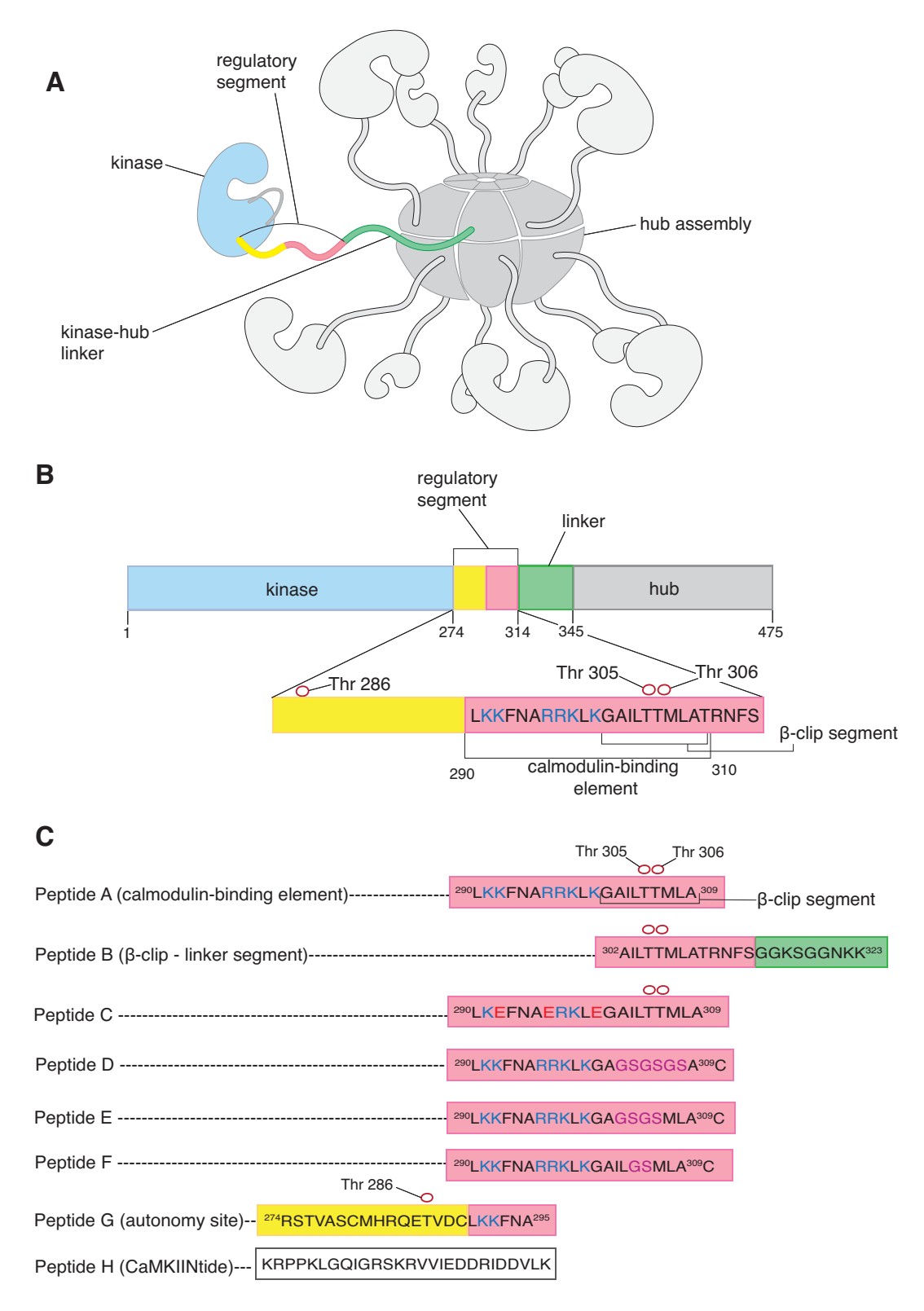

**Figure 1.** CaMKII architecture and peptides used in mass spectrometry analysis. (**A**) The architecture of dodecameric CaMKII, with subunits. The kinase domains are followed by the regulatory segments, which are connected to the hub domains by linkers. The hub domains (in darker grey) associate to form a ring-like hub assembly. For clarity, only a single kinase domain, regulatory segment, and linker are shown in colors, while the remainder are shown in shades of grey. In this subunit, the kinase domain is colored pale blue, the regulatory segment is colored yellow and pale pink, and the

*Figure 1 continued on next page*

*Figure 1 continued*

unstructured kinase-hub linker is colored green. (B) Domains of a CaMKII-α subunit. The color scheme is the same as in (A). The calmodulin-binding element and the β-clip segment are labeled. The positively charged residues of the calmodulin-binding element are colored deep blue. Phosphorylation sites are shown as empty red circles. (C) Peptides used in the mass spectrometry analyses. All the peptides are derived from the sequence of CaMKII-α, except Peptide H, a peptide that inhibits the kinase domain of CaMKII. The color schemes denote the region of the CaMKII-α subunit that the peptide is derived from and are the same as in (A). All the peptides, except Peptide H, are aligned with respect to Peptide A. Positively charged residues of the calmodulin-binding element are colored blue, and the β-clip segment is labeled in Peptide A. Phosphorylation sites in Peptide A, Peptide B, Peptide C and Peptide G are shown as empty red circles. In Peptide C, positively charged residues that are replaced with negatively charged residues are colored red. In Peptides D, E and F residues that are replaced with glycine and serine residues are colored purple.

## Results and discussion

### Mass spectrometry shows that peptides derived from the regulatory segment can break the CaMKII-α hub assembly

We used electrospray ionization mass spectrometry to analyze the integrity of the CaMKII-α hub assembly, in isolation and in the presence of peptides. The sequences of most of these peptides are based on that of the calmodulin-binding element of the regulatory segment (Peptide A and Peptides C-F, *Figure 1B*), which includes several positively charged residues preceding hydrophobic residues. We refer to the hydrophobic portion (residues 301–308) as the β-clip segment because a part of this segment is incorporated into the interfacial β-sheet in a structure of autoinhibited CaMKII-α, in a conformation referred to as the β-clip conformation (*Chao et al., 2011*). We also tested a peptide that contains only the β-clip segment and a portion of the kinase-hub linker (Peptide B). Finally, we tested the effect of two different peptides that bind to the kinase domain of CaMKII but have no known affinity for the hub. One of these peptides (Peptide G) corresponds to the portion of the regulatory segment that precedes the calmodulin-binding element (residues 274–295). This region of the regulatory segment binds to the kinase domain in the autoinhibited state and includes Thr 286, which confers $Ca^{2+}$/CaM-independent activity, or 'autonomy' when phosphorylated (*Lou and Schulman, 1989*). The other peptide (Peptide H) is a commonly used CaMKII inhibitor called CaMKIINtide, which binds to the kinase domain and blocks the active site (*Chang et al., 1998*; *Chao et al., 2010*).

The mass spectrometry experiments were carried out for the CaMKII-α hub in isolation, without the kinase domain and linkers. Intact CaMKII-α holoenzymes yield poor quality mass spectra due to sample heterogeneity resulting from proteolysis during the purification process (*Bhattacharyya et al., 2016*). The hub (120 μM subunit concentration) was incubated, with or without peptides, for 5 min at room temperature prior to injection into a time-of-flight mass spectrometer under soft ionization conditions. The resulting ions were detected, and the masses of the different species present in the sample were determined (see Materials and methods for complete description). The intact hub is detected in the *m/z* range of 5800 to 8000, and smaller oligomeric species appear in the *m/z* range of 2000 to 5200.

The mass spectrum of the CaMKII-α hub without added peptides is shown in *Figure 2A*. We observe a distribution of ions only in the higher *m/z* range, which correspond to dodecameric and tetradecameric hub assemblies. These observations are similar to the results of previous native mass spectrometry studies on isolated CaMKII-α hubs, in which dodecameric and tetradecameric hub assemblies were observed (*Bhattacharyya et al., 2016*; *McSpadden et al., 2019*). In the previous studies, a higher sample cone voltage of 150 V was used to reduce adduction of solvent and/or other small species, in order to obtain higher resolution mass spectra. Under these higher energy source conditions, some collision-induced disassembly of the hub occurs, resulting in the release of a small amount of monomeric species (*Figure 2—figure supplement 1A*). In the present study, we used softer ionization conditions, with a cone voltage of 50 V. Although this results in a decrease in the mass spectral resolution, we do not observe any smaller oligomeric species for the isolated hub. These acquisition conditions are therefore preferable for distinguishing between stable hub assemblies and those with peptide-induced disassembly, and were used for the remainder of the experiments.

Peptide A, containing the calmodulin-binding element, was added at final concentrations of 1 μM, 10 μM, 100 μM and 1 mM to CaMKII-α hub at a final subunit concentration of 120 μM. Following incubation, the sample was ionized and mass spectra were generated by averaging the scans for

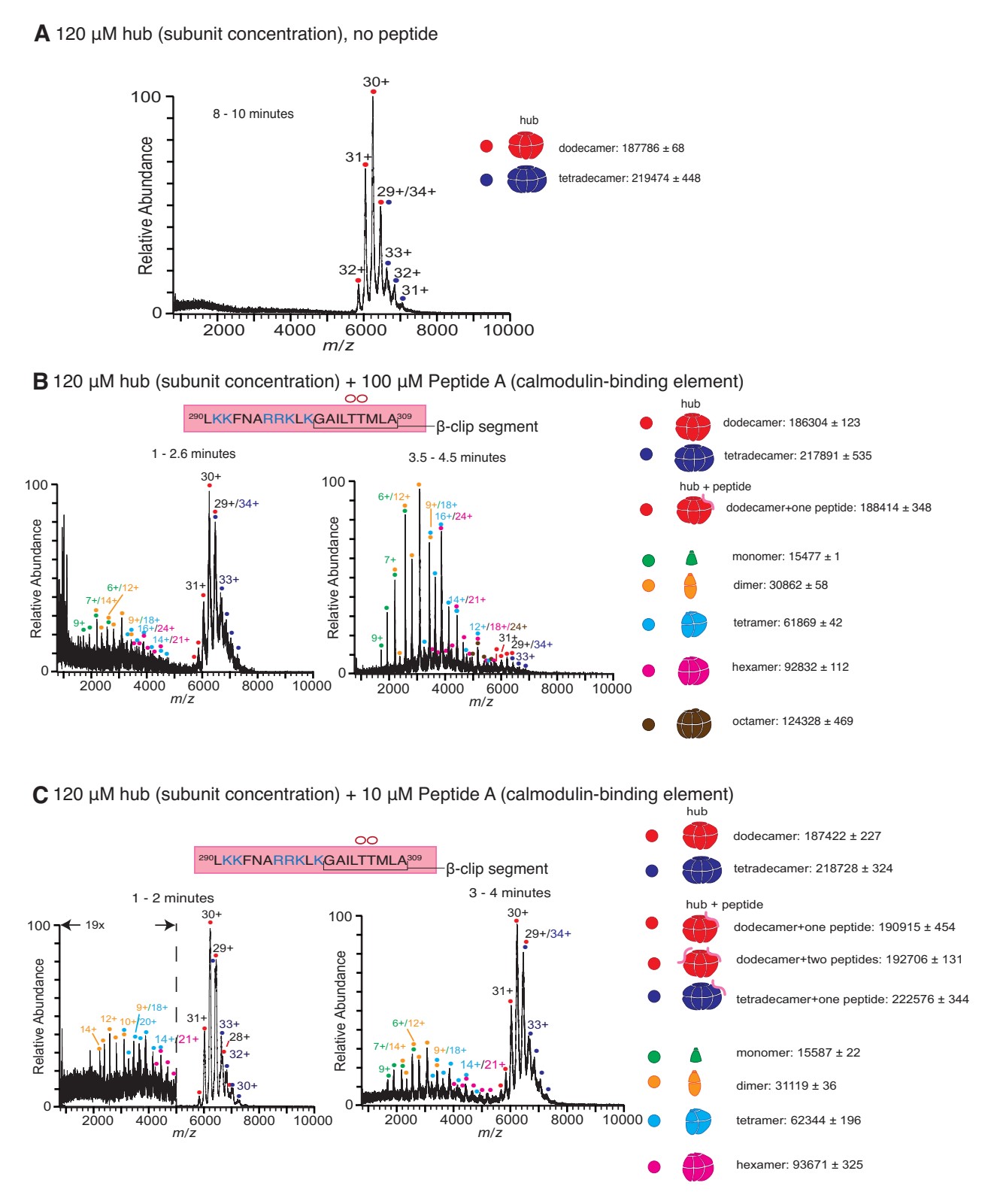

**Figure 2.** Native mass spectra of the hub incubated with peptides derived from the calmodulin-binding element of CaMKII-α. The masses (in Daltons) of the respective species are noted. (**A**) The isolated hub assembly is stable as dodecamers and tetradecamers in the absence of peptides. (**B**) The hub assembly (120 μM subunit concentration) incubated with 100 μM of Peptide A, shows disassembly within 3 min of initiation ionization (left). In addition to dodecamers and tetradecamers, monomers, dimers, tetramers, hexamers and octamers are observed. At ~ 4 min (right), the smaller oligomers are

*Figure 2 continued on next page*

*Figure 2 continued*

predominant. (**C**) The hub assembly (120 µM subunit concentration) incubated with 10 µM of Peptide A shows some release of smaller oligomeric species at ~2 min (left). This becomes more pronounced at 3–4 min (right), although the dodecameric and tetradecameric hub assemblies are also present. In all the spectra of the hub incubated with peptides, species corresponding to dodecameric or tetradecameric hub assemblies with one or two peptides bound are present.

The online version of this article includes the following figure supplement(s) for figure 2:

**Figure supplement 1.** Native mass spectra of the hub incubated with peptides derived from the calmodulin-binding element of CaMKII-α.

specific time segments after the initiation of electrospray ionization. The dissociation constant of Peptide A for the hub is ~90 µM (*Bhattacharyya et al., 2016*), and so we expect appreciable binding of the peptide to the hub for the condition with 100 µM of Peptide A. Indeed, the mass spectrometry results with 100 µM of Peptide A and the hub are dramatically different than for the hub alone (*Figure 2B*). In the presence of 100 µM of Peptide A, we observe hub species comprised of dodecamers and tetradecamers, as well as new species corresponding in mass to the intact complex with a peptide adducted. In addition, smaller oligomeric species that correspond to monomers, dimers, tetramers and hexamers of hub subunits appear within less than 3 min (*Figure 2B*). After 4 min, these smaller species are predominant.

Incubation of the hub with Peptide A at 10 µM concentration also yields spectra that demonstrate the release of smaller oligomeric species within ~ 1–2 min, although the peptide is sub-stoichiometric with respect to the hub under this mixing condition (*Figure 2C*). At ~ 3–4 min, the smaller species are prominent, although the dodecameric species are also present (*Figure 2C*). At even lower concentrations (1 µM) of Peptide A, we see essentially no disassembly of the hub, even after ~10 min, although there is some hexamer present at levels that are marginally above the noise. The abundance of peptide-adducted complex is significantly reduced at this lower peptide concentration. We note that under these conditions, the ratio of peptide to hub subunits is ~1:100 (*Figure 2—figure supplement 1B*) Incubation at saturating concentrations of peptide (1 mM) yields very noisy spectra with a high baseline consistent with significant protein/peptide aggregation (*Figure 2—figure supplement 1C*).

The incubation of the hub with 100 µM of Peptide B also causes disassembly of the hub and the release of monomers, dimers, tetramers and hexamers, but at a slower rate than seen with Peptide A. There are no smaller species observed for up to 2.5 min (*Figure 3A*); smaller oligomeric species are detectable only at ~3–4 min (*Figure 3A*). Peptide B does not contain the positively charged residues of the calmodulin-binding element. To test the importance of positive charge in Peptide A, we replaced three positively charged residues with glutamate (Peptide C, see *Figure 1B*). Addition of 1 mM of Peptide C to the hub did not cause hub disassembly (*Figure 3B*). These results are in agreement with previously reported binding measurements that showed that this peptide fails to bind to the hub (*Bhattacharyya et al., 2016*).

We replaced the β-clip segment of Peptide A with glycine and serine residues (Peptides D, E and F, see *Figure 1B*). Modification of the β-clip segment reduces or eliminates the ability of the peptides to disturb the integrity of the hub. The results of the incubation with 1 mM of Peptide D, where the entire β-clip segment is replaced with glycine and serine residues, are shown in *Figure 3C*. Only intact dodecamers and tetradecamers are observed after 8 min. Peptides E and F, in which only portions of the β-clip segment are replaced by glycine and serine residues, cause hub disassembly (*Figure 3—figure supplement 1A,1B*) but to a lesser extent than Peptide A at similar peptide concentrations (*Figure 2—figure supplement 1A*).

We also tested the effect of two control peptides that are known to interact with the kinase domain but are not known to bind to the hub. The first (Peptide G, *Figure 1B*) contains the autonomy site (Thr 286) of the regulatory segment (*Lou and Schulman, 1989*), and the second, CaMKIINtide (Peptide H, *Figure 1B*), is an inhibitor that binds to the substrate recognition site of the kinase domain (*Chang et al., 1998*; *Chao et al., 2010*). Some adduction of these peptides to the intact complexes is observed, but neither peptide induces hub disassembly (*Figure 3D*, *Figure 3—figure supplement 1C*), pointing to a specific role for the calmodulin-binding element in disturbing the integrity of the CaMKII-α hub.

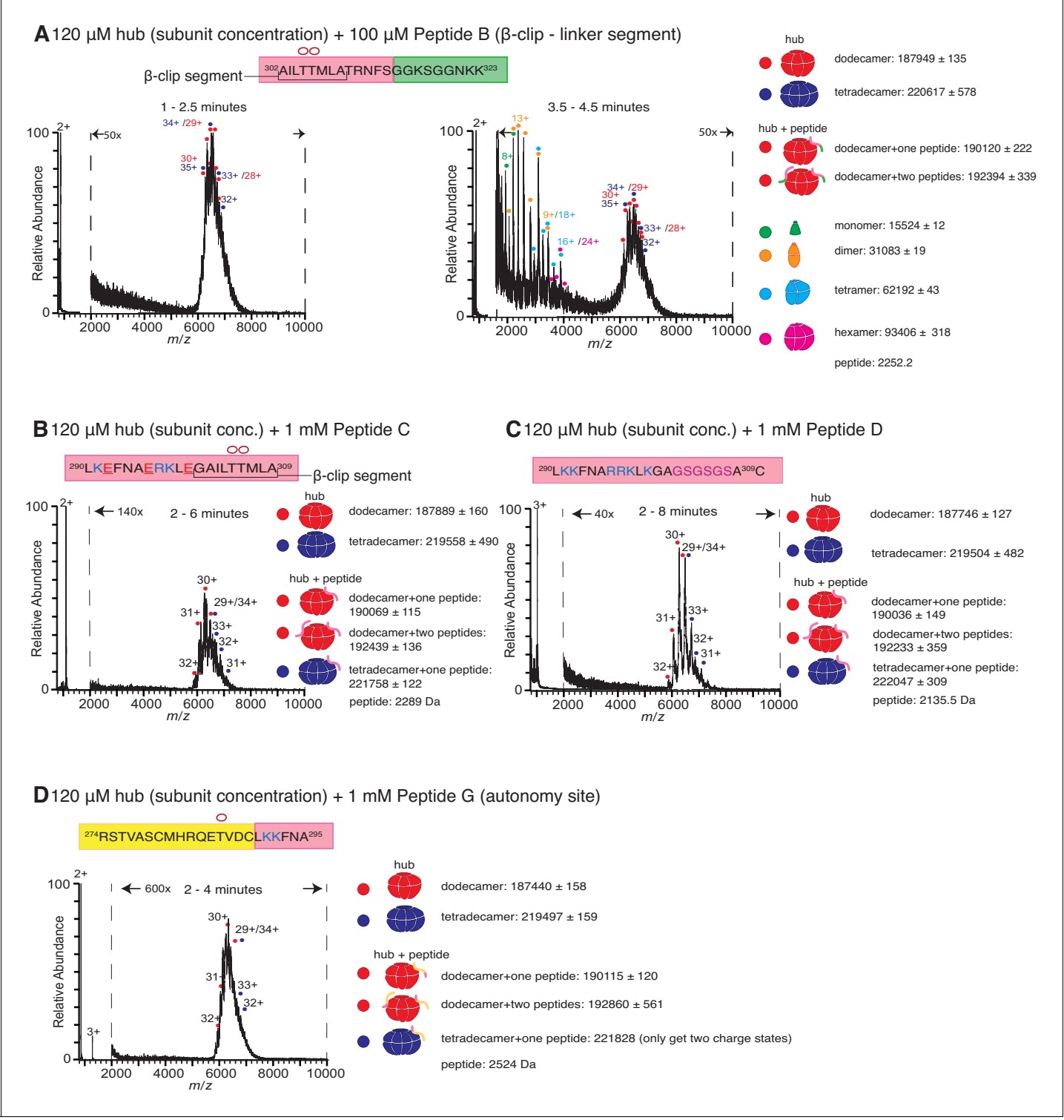

**Figure 3.** Role of different regions of the calmodulin-binding element of CaMKII-α in inducing hub disassembly. The masses (in Daltons) of the respective species are noted. (**A**) Peptide B, which contains the β-clip segment of the calmodulin-binding element, but not the positively charged residues, induces disassembly, but at a slower rate than Peptide A. When 100 μM of Peptide B is incubated with the hub assembly, no smaller species are observed up to ~2.5 min after initiation of ionization (left). At 3.5–4.5 min, smaller oligomeric species corresponding to monomers, dimers, tetramers and hexamers are observed (right). (**B**) Incubation of the hub assembly with 1 mM of Peptide C, in which three positively charged residues of Peptide A are replaced with negatively charged residues (colored red in sequence), does not result in the release of smaller oligomeric species up to 6 min after ionization, even though the peptide binds to the hub assembly. (**C**) Modifying the β-clip segment of the calmodulin-binding element

*Figure 3 continued on next page*

*Figure 3 continued*

abrogates disassembly. Incubation of the hub assembly with 1 mM of Peptide D, in which the β-clip segment of Peptide A is replaced with glycine and serine residues (colored purple) does not result in the release of smaller oligomeric species, and only dodecamers and tetradecamers are observed up to 8 min after ionization. (D) Incubation of the hub assembly with 1 mM of Peptide G, whose sequence corresponds to the autonomy site of the CaMKII-α regulatory segment, does not result in the release of smaller oligomeric species up to 4 min after ionization.

The online version of this article includes the following figure supplement(s) for figure 3:

**Figure supplement 1.** Role of different regions of the calmodulin-binding element of CaMKII-α in inducing hub disassembly.

The calmodulin-binding element contains two residues, Thr 305 and Thr 306, that can be phosphorylated when CaMKII is activated, and phosphorylation at these sites prevents rebinding of $Ca^{2+}$/CaM to the holoenzyme (*Colbran, 1993*; *Bhattacharyya et al., 2020b*). We tested the stability of the hub in the presence of variants of Peptides A and B (*Figure 1C*) in which Thr 305 is phosphorylated (hereafter referred to as Peptide $A^{phos}$ and Peptide $B^{phos}$, respectively). In these experiments, CaMKII-α hub at a final subunit concentration of 120 μM was incubated with Peptide $A^{phos}$ for 5 min at room temperature prior to injection into the mass spectrometer under similar ionization conditions as before. Under these conditions, the mass spectra obtained comprised predominantly of smaller oligomeric species and aggregates, suggesting that the breakage of the hub is accelerated in the presence of the phosphorylated peptide compared to the unphosphorylated peptide. We dispensed with the incubation step, and the sample was injected directly into the mass spectrometer as soon as the hub, at a final subunit concentration of 120 μM, was mixed with Peptide $A^{phos}$ at final concentrations of 1 μM, 10 μM and 100 μM (*Figure 4*). The hub undergoes disassembly in the presence of Peptide $A^{phos}$ to release hexamers, tetramers, dimers and monomers. At the higher peptide concentrations of 100 μM and 10 μM of Peptide $A^{phos}$, smaller oligomeric species appear within ~1 min (*Figure 4A,B*). At ~2–3 min, the spectra of the hub incubated with Peptide $A^{phos}$ at these higher concentrations are very noisy, consistent with significant protein and peptide aggregation. At a lower concentration of 1 μM of Peptide $A^{phos}$, smaller oligomeric species do not appear until ~2 min after ionization *Figure 4C*. We also tested the stability of the hub (120 μM final subunit concentration) when mixed with 100 μM of Peptide $B^{phos}$, again with no pre-incubation of the peptide with the hub, and observed smaller oligomeric species within a minute (*Figure 4D*). These smaller species are predominant after ~1.5 min.

The data show that phosphorylated versions of the regulatory segments are more effective at destabilizing the hub than the unphosphorylated species (compare the data shown in *Figures 2* and *3A* with the data in *Figure 4*). This result is unexpected, since the phosphorylated and unphosphorylated peptides bind to the hub with comparable affinity (*Bhattacharyya et al., 2016*). Based on the available crystal structures of the hub, an explanation for why the phosphorylated peptides are more potent at breaking the hub is not readily apparent. Also, we note that the phosphorylated peptides do not appear to provide a persistent signal in the mass spectrometer, and over time the peak corresponding to the peptide decreases and eventually disappears. For these reasons, the interaction of the phosphorylated peptides with the hub requires further study in order to obtain a more complete understanding of their mechanism of action.

The mass spectrometric data presented here provide compelling evidence that the interaction of the hub with the calmodulin-binding element of CaMKII, whether phosphorylated or not, results in hub destabilization. Interestingly, this destabilization is not detected in conventional experiments using gel filtration. For example, gel filtration analysis of the CaMKII hub incubated with Peptide A under conditions identical to those used for mass spectrometry shows only a single protein peak corresponding to dodecameric/tetradecameric species (*Figure 4—figure supplement 1*). Since Peptide A interacts with the hub relatively weakly (*Bhattacharyya et al., 2016*), it is possible that gel filtration causes dissociation of the peptide from the hub, perhaps allowing the hub to reassemble. In addition, perturbations introduced by the mass spectrometric analysis, such as sample heating, might partially destabilize the hub making it possible to more readily observe the effects of the peptides on the hub stability.

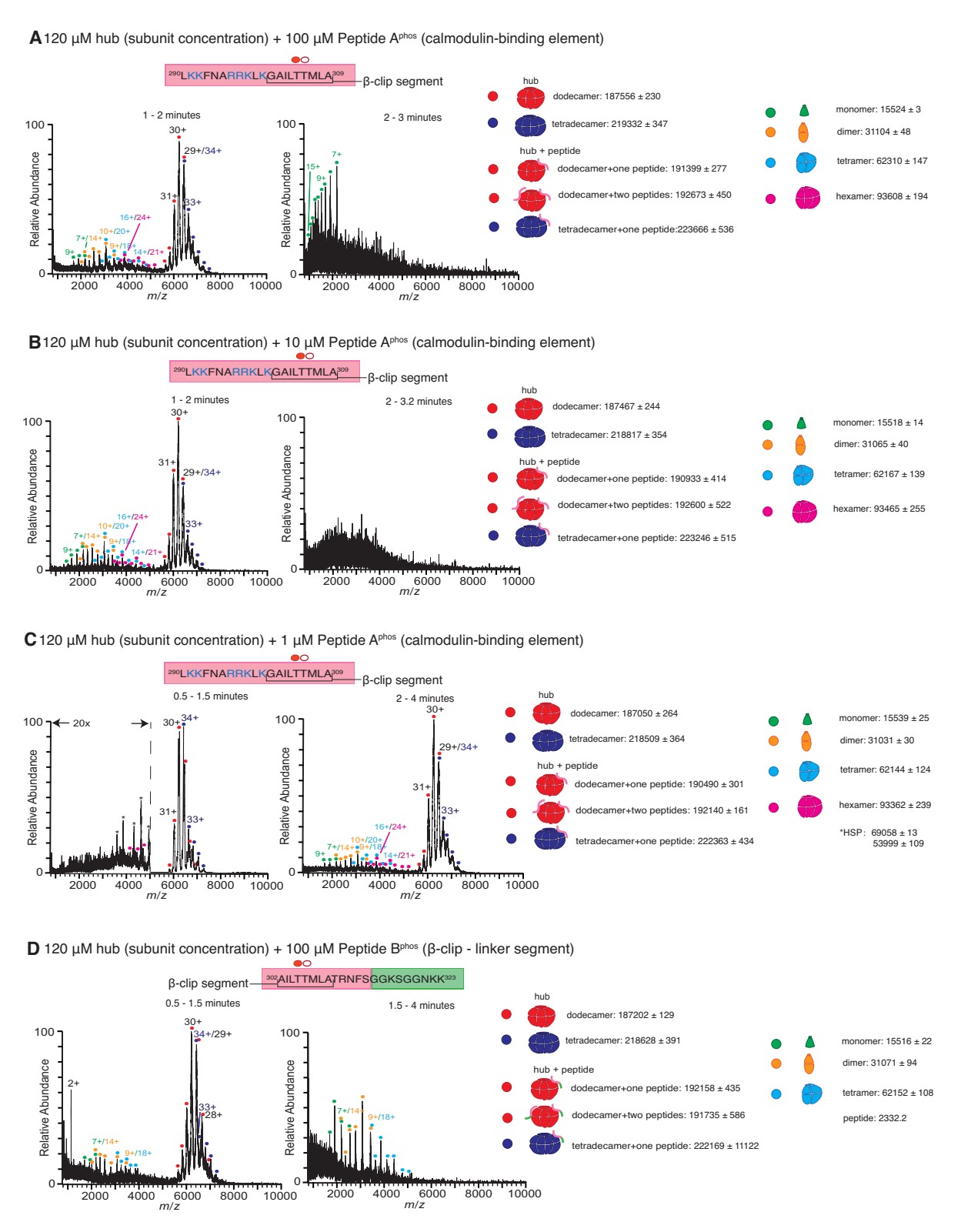

**Figure 4.** Native mass spectra of the hub incubated with phosphorylated peptides derived from the calmodulin-binding element of CaMKII-α. Due to the rapid rate at which the hub is destabilized in the presence of phosphorylated peptide, the 5-min incubation step was dispensed which and the sample was injected into the mass spectrometer as soon as the hub was mixed with the peptide. The phosphorylated residue (Thr 305) is indicated with a filled red circle. The masses (in Daltons) of the respective species are noted. (**A**) The hub assembly (120 μM subunit concentration) incubated with 100

*Figure 4 continued*

µM of Peptide A^phos shows disassembly within 1–2 min after the start of ionization (left). In addition to dodecamers and tetradecamers, monomers, dimers, tetramers and hexamers are observed. At ~2–3 min (right), the spectra consist primarily of protein and peptide aggregates. (**B**) The hub assembly (120 µM subunit concentration) incubated with 10 µM of Peptide A^phos also shows disassembly within 1–2 min of ionization (left). In addition to dodecamers and tetradecamers, monomers, dimers, tetramers and hexamers are observed. At ~2–3 min (right), the spectra consist primarily of protein and peptide aggregates. (**C**) The hub assembly (120 µM subunit concentration) incubated with 1 µM of Peptide A^phos shows almost no disassembly up to ~1.5 min (left). At ~2–4 min smaller species including hexamers, tetramers, dimers and monomers are observed, in addition to dodecamers and tetradecamers. (**D**) The hub assembly (120 µM subunit concentration) incubated with 100 µM of Peptide B^phos shows some disassembly into tetramers, dimers and monomers within ~1.5 min of initiation of ionization (left). At ~1.5–4 min these smaller species are predominant. The online version of this article includes the following source data and figure supplement(s) for figure 4:

**Source data 1.** Numerical data plotted in supplement accompanying *Figure 4*.
**Figure supplement 1.** Analytical gel filtration of the CaMKII-α hub with and without Peptide A.

## Long-timescale molecular dynamics simulations of regulatory segment docking on the hub

We generated molecular dynamics trajectories for the hub assembly with the regulatory segments and linkers either present or absent, using the specialized Anton2 supercomputer (*Shaw et al., 2014*). We built a dodecameric hub assembly of CaMKII-α, based on the crystal structure of the CaMKII-α hub (PDB ID – 5IG3) (*Bhattacharyya et al., 2016*), on to which we modeled the unstructured linker segments (residues 311–345 in CaMKII-α) and the regulatory segments (residues 281–310). The regulatory segments and linkers were modeled in arbitrary conformations, and located at variable distances from the surface of the dodecameric hub assembly. A 13 µs molecular dynamics trajectory was generated for this system. We also generated a 6 µs trajectory of the hub assembly without the linkers and regulatory segments. The lengths of the trajectories were limited by the availability of time on the Anton2 supercomputer.

In the simulation with the regulatory segments present, two of the twelve regulatory segments dock on the hub (*Figure 5*, *Figure 5—figure supplement 1B*). Using the subunit notation shown in *Figure 5—figure supplement 1A*, these are the regulatory segments of subunits F and L, which dock at the interfaces between subunits B and L, and subunits F and H, respectively (*Figure 5*). The docking involves hydrogen-bond formation between the calmodulin-binding element (residues 292–310) and the open edge of the β-sheet (residues 410–416) located at the interface (*Figure 5—figure supplement 2A,2B*). The regulatory segment of subunit L docks within 0.5 µs (hereafter referred to as Docking 1), and that of subunit F docks at ~3 µs (referred to as Docking 2) (*Figure 5—figure supplement 1B*).

Both docking events are initiated by interactions between positively-charged residues on the regulatory segments and negatively-charged residues that line the interfacial groove (*Figure 5—figure supplement 1C*), although there are differences in the specific interactions that are made and in the register of the regulatory segments within the interfacial grooves. Backbone hydrogen bonds are then formed between the regulatory segment and the β-sheet at the interface, leading to the incorporation of this portion of the regulatory segment into the β-sheet, similar to the β-clip (*Chao et al., 2011*). The hydrophobic sidechains of the β-clip segment pack into the hydrophobic core of the subunits, further stabilizing docking by the regulatory segment. Unlike the electrostatic interactions, which are transient, the backbone hydrogen bonds are stable and persist for the duration of the simulation (*Figure 5—figure supplement 1C*).

The interfaces at which two dockings occur are located diametrically across from each other on the hub (*Figure 5—figure supplement 1A*) and docking is accompanied by a large conformational change at both interfaces (*Figure 6—figure supplement 1A*). We quantified this distortion by measuring the angle between the axes of the N-terminal α-helices (the αA helices) of two adjacent subunits (*Figure 6A*). The core structures of the individual subunits remain relatively unchanged over the course of the simulation (*Figure 6—figure supplement 2A*), allowing us to use the angle between the axes of the αA helices as a measure of the overall rotation of one subunit at an interface with respect to the other. The initial value of the inter-helix angle between the αA helices of subunits B and L, which bracket the interface at which Docking 1 occurs, is ~50°. These subunits begin to rotate away from each other when the trajectory is initiated, and the angle between helices αA has increased to ~60° when Docking 1 is initiated (*Figure 6C*). At this point, there is a marked increase in

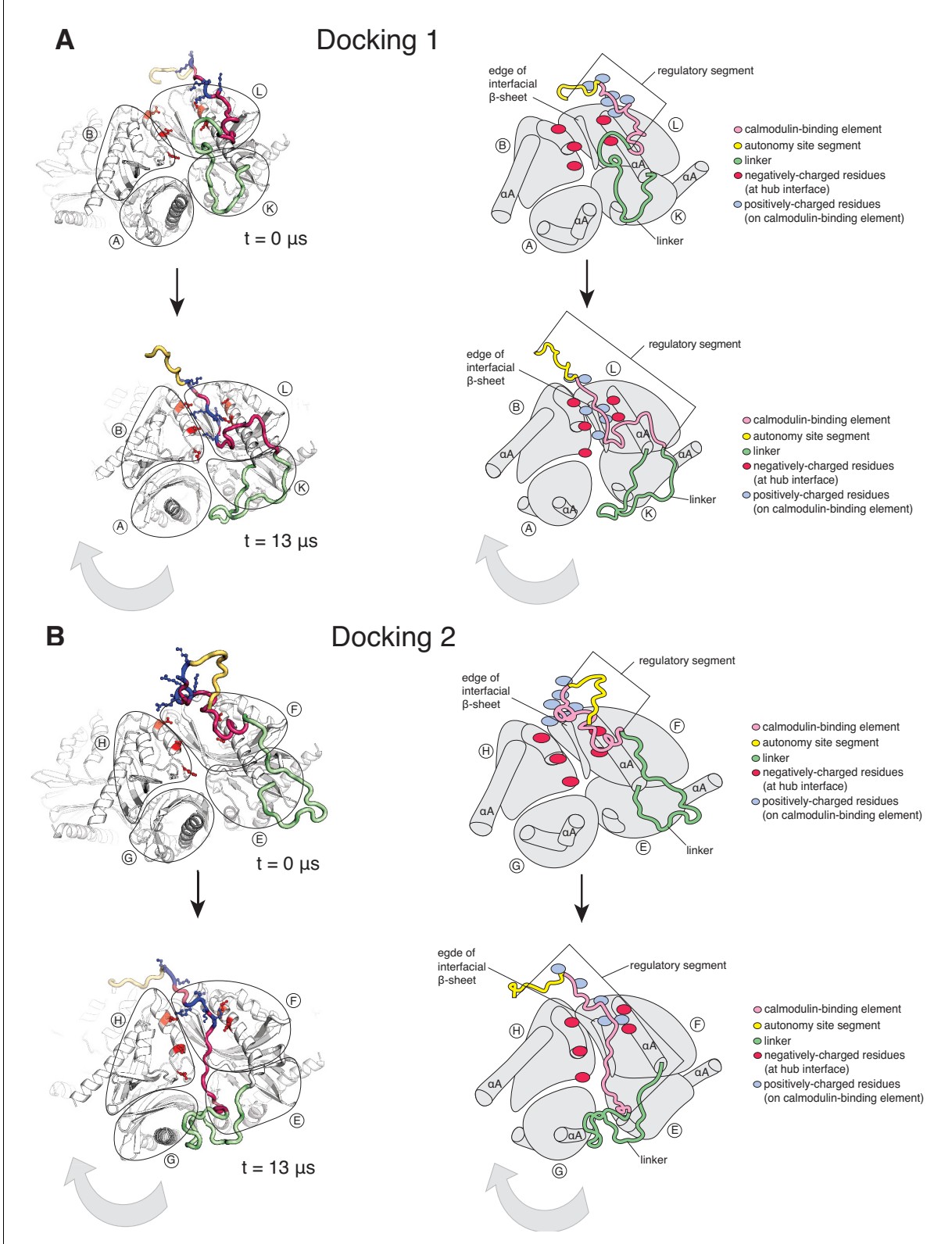

**Figure 5.** Calmodulin-binding elements from two of the twelve regulatory segments dock on the hub in the simulation of the dodecameric CaMKII-α hub, with considerable distortions at the interfaces where they bind. The color scheme for the regions of the subunits are the same as in *Figure 1A*. (A) Instantaneous snapshot from the start (top) and end (bottom) of the simulation showing interface where Docking 1 occurs. The calmodulin-binding element of subunit L docks at the interface between the hub domains of subunits B and L. The positively-charged residues of the regulatory segment

*Figure 5 continued on next page*

*Figure 5 continued*

(shown in blue) form interactions with the negatively-charged residues (shown in red) that line the interface between subunits B and L. The hub domain of subunit B rotates away from the hub domain of subunit L. (**B**) Instantaneous snapshot from the start (top) and end (bottom) of the simulation showing interface where Docking 2 occurs. The calmodulin-binding element of subunit F docks at the interface between the hub domains of subunits F and H. The positively-charged residues of the regulatory segment (shown in blue) form interactions with the negatively-charged residues (shown in red) that line the interface between subunits F and H. The hub domain of subunit H rotates away from the hub domain of subunit F.

The online version of this article includes the following source data and figure supplement(s) for figure 5:

**Source data 1.** Numerical data plotted in supplement accompanying *Figure 5*.
**Figure supplement 1.** Docking of regulatory segments onto hub assembly.
**Figure supplement 2.** Close-up of two docking events in the simulation, in comparison with a crystal structure where a peptide docks at the interface.

the angle between the αA helices (to ~80°), where it remains for the rest of the trajectory (*Figure 6B*).

Subunits F and H, which bracket the interface where Docking 2 occurs, also rotate away from each other from the start of the simulation (*Figure 6B*). The angle between the αA helices at this interface increases from an initial value of ~50° to ~60° by 3 μs, when Docking 2 is initiated (*Figure 6C*), and then continues to increase over the rest of the simulation (*Figure 6B*). During the course of the simulation, the hub distorts from its normal circular shape to an oval shape, but it does not actually separate at any of the interfaces (*Figure 6—figure supplement 1B D*). This is because the large distortions at the two docking interfaces are compensated for by closure of the remaining interfaces (*Figure 6*, *Figure 6—figure supplement 2B*). Hydrogen bonds across the subunit interfaces are maintained, irrespective of whether an interface opens or closes (*Figure 6—figure supplement 1B D*). Nevertheless, given the large distortions in the geometry of the hub with respect to the crystal structure, the hub assembly may be destabilized as a consequence of the binding of just one or two regulatory segments. Alternatively, more than one regulatory segment might have to bind to promote disintegration of the hub. In the simulation, at interfaces where regulatory segments are not bound the interfacial grooves are constricted so that binding of the regulatory segments would be hindered substantially (*Figure 6—figure supplement 3*). This suggests that the binding of more than two regulatory segments to the hub would have to be accompanied by additional distortions and a loss of integrity of the hub assembly.

The rotation between the subunits that bracket the docking interfaces is much larger in the simulation than that observed in any crystal structures of the hub, including those with peptides bound at interfaces (*Bhattacharyya et al., 2016*; *Chao et al., 2011*). In a structure of the hub assembly of CaMKII from the sea anemone *Nematostella vectensis*, the N-terminal portion of the linker preceding the hub folds back to interact with the hub (*Bhattacharyya et al., 2016*; *Figure 5—figure supplement 2C*). In this structure, as in the simulation, the linkers are incorporated as an additional strand of the interfacial β-sheet (*Figure 5—figure supplement 2C*). The *N. vectensis* hub assembly is slightly cracked, and it assumes a spiral, lock-washer conformation (*Bhattacharyya et al., 2016*). The interaction between the linker and the *N. vectensis* hub involves acidic residues in the linker interacting with acidic residues in the hub. This interaction is likely to be an artefact of the low pH of crystallization and neutralization of acidic residues (*Bhattacharyya et al., 2016*). In contrast, the interaction between the regulatory segment and the hub seen in the simulation involves favorable interactions between oppositely charged sidechains.

## Regulatory segment docking appears to trap an intrinsic mode of distortion of the hub

It is striking that in the simulation a very similar distortion is seen at both docking sites, even though the register of the regulatory segment at one inter-subunit groove is offset by four residues with respect to the docking at the other inter-subunit groove (*Figure 5—figure supplement 2C*). This suggests that the distortion at the two docking sites reflects an intrinsic feature of the hub dynamics, one that does not depend on the details of the interaction with the peptide. In a 6 μs simulation of the hub without the regulatory segments, the hub assembly showed large structural excursions at interfaces between hub domains, including ones that are similar to those observed in the simulation of the hub with regulatory segments (*Figure 6—figure supplement 2B*). Large displacements of

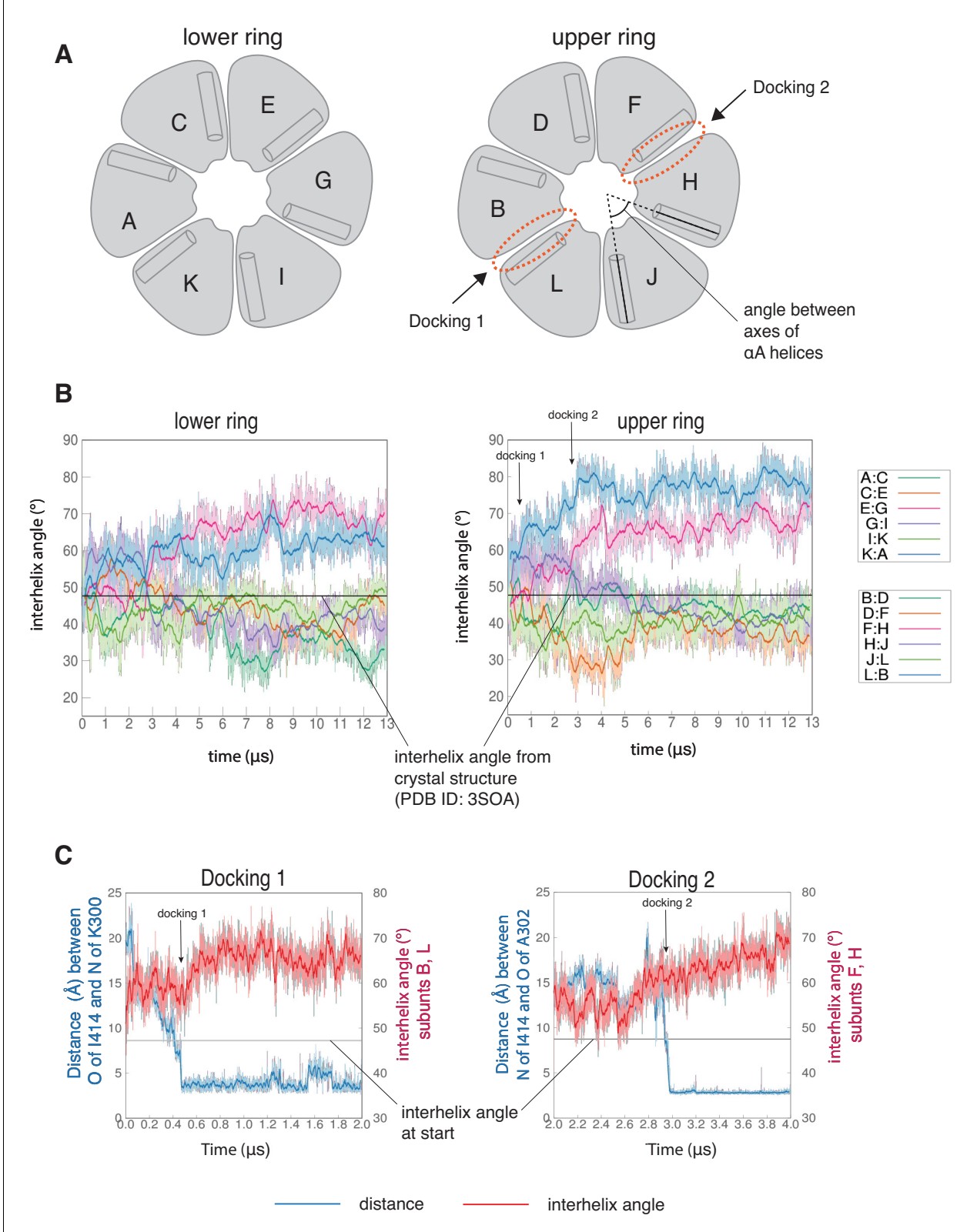

**Figure 6.** Interfaces at which docking occur are stabilized in an open form, while remaining interfaces close. (**A**) Schematic showing the naming scheme used to identify subunits. The interhelix angle between axes of αA helices are used to measure the rotation of each hub domain with respect to the adjacent hub domain. (**B**) Variation of the angle between the axes of the αA helices of adjacent subunits in the simulation with the regulatory segment. The darker traces are the time-averaged values of the interhelix angles calculated using a moving window of 240 ns, while the lighter shades are the

*Figure 6 continued on next page*

*Figure 6 continued*

actual distances. The two interfaces where regulatory segments dock between subunits L and B (colored blue), and between subunits F and H (colored pink) open and the remaining interfaces close. In the lower ring the interfaces mirror the behavior of the interfaces in the upper ring. The interhelix angle between the αA helices of adjacent subunits from the crystal structure of an autoinhibited, dodecameric holoenzyme (PDB ID - 3SOA) (*Chao et al., 2011*), which is perfectly symmetrical, is indicated by the horizontal black line. (C) Variation in the angle between the axes of the αA helices of hub domains that flank the interfaces where the calmodulin-binding elements dock (shown in red), and the distance between a residue from the calmodulin-binding element and the interfacial β-sheet (shown in blue), over the course of the simulation. The darker traces are the time-averaged values, calculated using a moving window of 12 ns, while the lighter shades are the actual distances. The interhelix angle at the start of the simulation is indicated by a horizontal grey line.

The online version of this article includes the following source data and figure supplement(s) for figure 6:

**Source data 1.** Numerical data plotted in *Figure 6* and accompanying figure supplement.
**Figure supplement 1.** Close-up of hub interfaces from the molecular dynamics simulation of CaMKII-α.
**Figure supplement 2.** Individual subunits of the dodecameric hub are not very dynamic, while the interfaces between the subunits are intrinsically dynamic.
**Figure supplement 3.** Model of regulatory segment docked onto the interface between subunits L and J, which closes over the course of the simulation.

subunits were also seen in a short 100 ns simulation of the hub reported previously (*Bhattacharyya et al., 2016*).

To better understand the intrinsic dynamics of the hub assembly, we used normal mode analysis. We determined the normal modes of a dodecameric hub assembly (*Chao et al., 2011*), without the linkers, regulatory segments and the kinase domains, using the *ElNémo* server (*Suhre and Sanejouand, 2004a*; *Suhre and Sanejouand, 2004b*). This server uses the Elastic Network Model (*Tirion, 1996*) and the rotation–translation block method (*Durand et al., 1994*; *Tama et al., 2000*) to compute the normal modes.

To analyze the normal modes, we generated pairs of structures representing the sweep of each normal mode. Such pairs of structures were compared to the initial and final structures from the molecular dynamics trajectory. For each comparison, one of the two subunits at an interface was aligned between the pairs of structures, and the deviation in Cα positions of the adjacent subunit was considered. This gives us a measure of the differences in orientations of adjacent subunits between the normal mode displacement and the molecular dynamics trajectory.

We observed a striking similarity between the displacement seen in the molecular dynamics simulation and that seen in the lowest-frequency internal modes (*Figure 7A,B*). In these normal modes, just as in the molecular dynamics simulation, two of the inter-subunit interfaces open, and the other interfaces close. Thus, it appears that the coupled opening of two interfaces and the correlated closing of the others is an intrinsic property of the hub assembly. In the molecular dynamics simulation, it appears that the regulatory segments take advantage of the opening of the interfaces to form stable interactions with the interfaces and trap this distorted structure.

## Activation-induced destabilization of CaMKII-α holoenzymes expressed in mammalian cells

We studied the effect of activation on the stability of the intact CaMKII-α holoenzyme expressed in mammalian cells, using TIRF microscopy and single-molecule analysis. We developed a cell-based assay that allowed us to rapidly isolate CaMKII-α after cell lysis, while reducing the heterogeneity that can result from proteolysis or the aggregation that occurs during purification (*Bhattacharyya et al., 2020b*). Briefly, CaMKII-α was tagged at the N-terminus with a biotinylation tag and mEGFP (*Cormack et al., 1996*; *Zacharias et al., 2002*) and overexpressed in HEK293T cells. The cells were lysed, and CaMKII-α was pulled down from the diluted cell-lysate onto a streptavidin-coated glass coverslip (*Figure 8A*). CaMKII-α is visualized on the glass slide as spots of fluorescence intensity in the GFP channel at 488 nm.

The distribution of fluorescence intensities for all the samples has two major peaks, one at lower intensities and one at higher intensities (*Figure 8B*). Additionally, there is a shoulder in the lower intensity peaks in wild-type mEGFP-CaMKII-α (*Figure 8B*). Step photobleaching analyses and a comparison with the intensity distribution of mEGFP-Hck, a monomeric protein kinase, indicate that the peaks with lower fluorescence intensity correspond to CaMKII-α monomers, while the shoulder

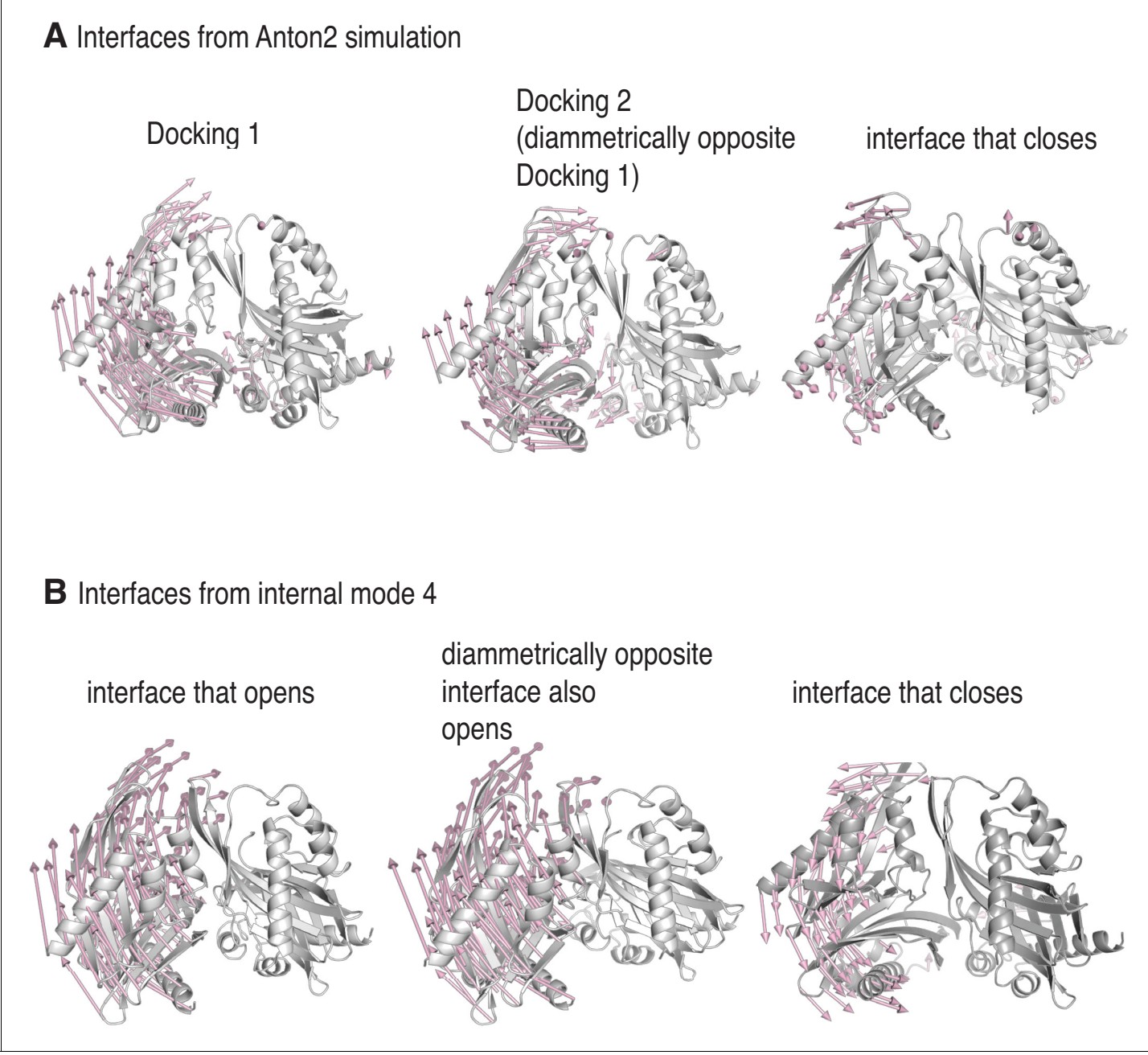

**Figure 7.** Docking traps the hub assembly in a conformation similar to one of its low-frequency normal modes. (A) Conformational distortion in interfaces from the molecular dynamics simulation. The cartoon representations show the interfaces at the start of the simulation. Vectors (in pale pink) indicate the movement of the Cα-atoms of the hub domains to the left of the interfaces, with respect to the hub domains to the right of the interfaces, over the course the simulation. The interface at which Docking 1 (left) and Docking 2 (center) occur, open. One of the interfaces that closes is shown (right), and in this the hub domains to the left of the interface moves inwards, toward the interface. For clarity, only vectors from every other Cα-atom are shown. (B) Conformational change of interfaces in internal mode four from the normal mode analysis. The vectors (in pale pink) indicate the movement of the Cα-atoms of the hub domains to the left of the interfaces, with respect to the hub domains to the right of the interfaces, along the vector. Two of the interfaces (left and center) in internal mode four open, just as in the simulation, while the remainder (representative interface shown on right) close. For clarity, only vectors from every other Cα-atom are shown.

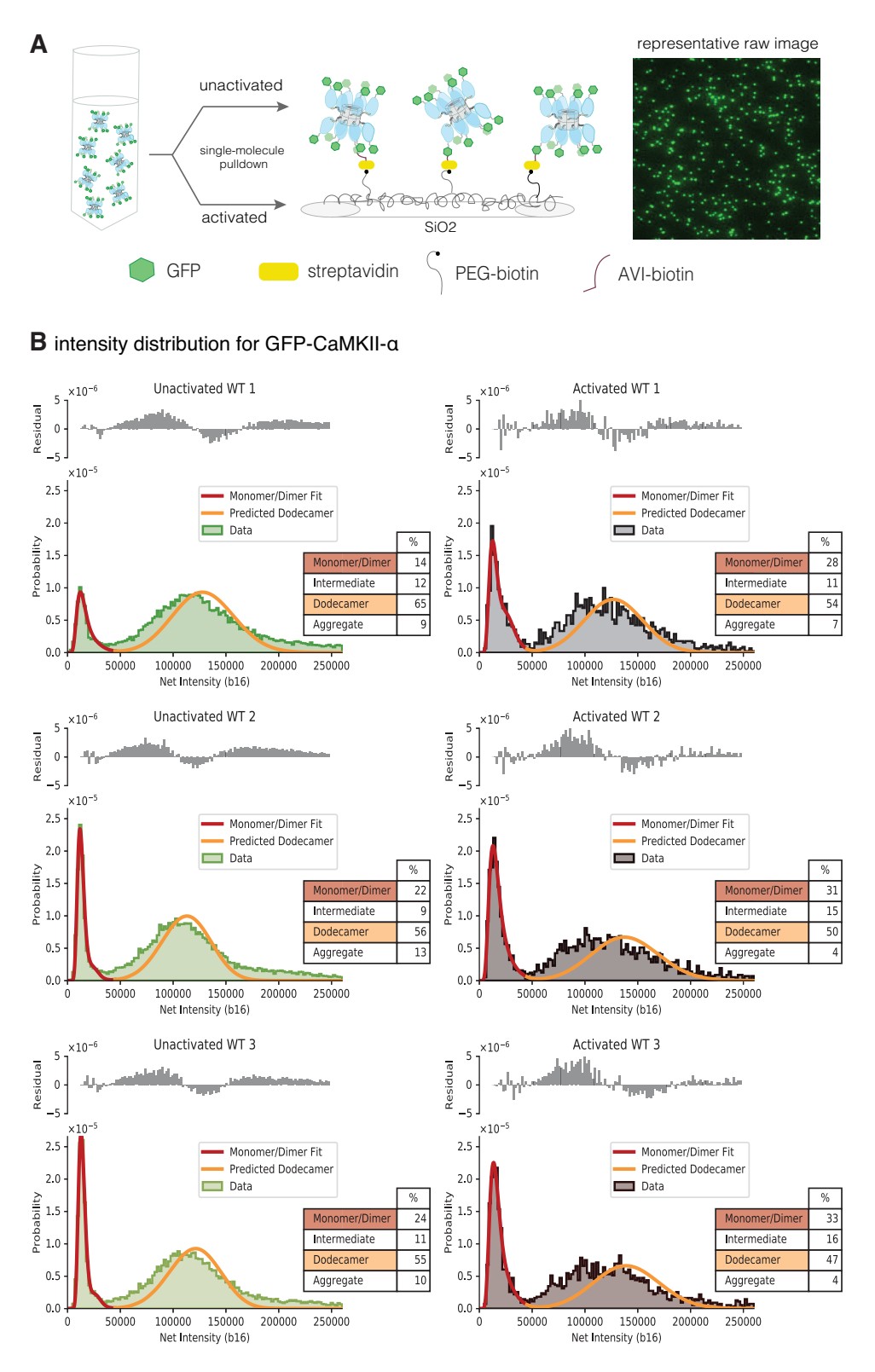

**Figure 8.** Activation destabilizes the CaMKII-α holoenzyme. (**A**) Schematic showing the experimental setup. Biotinylated mEGFP-CaMKII-α was overexpressed in HEK 293 T cells, the cells were lysed and mEGFP-CaMKII-α from the diluted cell lysate was either first activated, or directly pulled down onto a glass coverslip, functionalized with streptavidin, for visualization at single-molecule resolution. The fluorescence intensity of each individual mEGFP-CaMKII-α is correlated with its oligomeric state. (**B**) Distribution of the intensities for three replicates of unactivated (left) and activated (right)

*Figure 8 continued on next page*

*Figure 8 continued*

mEGFP-CaMKII-α. Two major peaks are observed, with a peak at lower intensity, and a peak at higher intensity. There is a shoulder in the peak at lower intensity, and together these correspond to a mixture of monomers and dimers, with the dimer intensities occurring in the shoulder. The peak at higher intensities corresponds to intact dodecameric holoenzymes. The intensity data were fit to a mixture comprising of monomers and dimers (red line) and the position of the peak corresponding to dodecamers was predicted (orange line). The majority of the area of the histogram is captured by these three components; however, there are oligomers of intermediate sizes also present. The residual plot shows the deviation from data with the fitted three-component distribution and shows the intermediate oligomeric species (between dimer and dodecamer). The percentage of the smaller, intermediate, dodecameric and aggregate species in the samples, as estimated from the fit are shown. Upon activation, the area of the peak at lower intensity rises, with a ~1.3–2-fold increase across three replicates, indicating that upon activation the holoenzyme undergoes some disassembly to form dimers. The percentage of intermediate oligomeric species also increases upon activation.

The online version of this article includes the following source data, source code and figure supplement(s) for figure 8:

**Source data 1.** Numerical data plotted in *Figure 8* and accompanying figure supplement.
**Source code 1.** Source code used to generate figures plotted in *Figure 8* and *Figure 9*.
**Figure supplement 1.** Peaks at lower intensities correspond to monomeric and dimeric species of CaMKII-α.

corresponds to CaMKII-α dimers (*Figure 8—figure supplement 1A,1B*). Some percentage of GFP is dark and so a fraction of the spots with intensities corresponding to monomeric species actually correspond to dimeric species. For spots with higher fluorescence intensity in the CaMKII-α samples, photobleaching does not result in clearly resolvable steps (*Figure 8—figure supplement 1A*).

We estimated the number of subunits that comprise the peak at higher intensity by adapting the method described by *Mutch et al., 2007*, where the position of the intensity peak for an oligomer with N subunits can be determined by convolving the monomeric (log-normal) distribution peak with itself N times. We also estimated the contribution of dark GFP and accounted for this in our calculation of the position of the N-oligomer peak (see Materials and methods for more details). Approximately 30% of the GFP across all samples was found to be dark (*Table 1*), consistent with previous findings (*Ulbrich and Isacoff, 2007*). Here, we fit the peak at lower intensity to a model comprised of monomeric and dimeric species. We extracted the parameters for the underlying monomeric log-normal distribution from this fit and convolved it with itself 12 times, while accounting for dark EGFP, to predict the location and width of a peak comprising of dodecameric species. The predicted dodecamer intensity distribution corresponds well with the experimentally observed fluorescence intensity distribution of mEGFP-CaMKII-α (*Figure 8B*), indicating that this peak corresponds to predominantly dodecameric holoenzymes. From the fit, we were able to determine the percentage of smaller (monomeric and dimeric) species and larger, dodecameric species in the sample (*Figure 8B*).

For unactivated mEGFP-CaMKII-α, the intensity distributions indicate the presence of a small population of monomers and dimers and a larger fraction of intact dodecameric holoenzymes (*Figure 8B*). We activated CaMKII-α in the diluted cell lysate with saturating amounts of $Ca^{2+}$/CaM (5 μM) and ATP (10 mM), captured the activated CaMKII-α on a coverslip functionalized with streptavidin, and measured the distribution of fluorescence intensity of the imaged spots. These conditions have been shown previously to result in robust phosphorylation of CaMKII-α on Thr 286, with a lower amount of phosphorylation on Thr 305/Thr 306 (*Bhattacharyya et al., 2020b*).

**Table 1.** Estimated fraction of dark GFP.

|  | NUs |
| --- | --- |
| Unactivated WT 1 | 0.282 |
| Activated WT 1 | 0.279 |
| Unactivated WT 2 | 0.328 |
| Activated WT 2 | 0.287 |
| Unactivated WT 3 | 0.296 |
| Activated WT 3 | 0.299 |
| Unactivated FA | 0.274 |
| Activated FA | 0.332 |

The fluorescence intensity distribution for CaMKII-α, along with the modeled distribution and the estimated percentages of the respective oligomeric species are shown in *Figure 8B*. There is a shift in the intensity distribution curve when compared to unactivated CaMKII-α, with the peak that corresponds to smaller (monomeric and dimeric) CaMKII-α showing a ~1.3–2-fold increase in peak area, indicating that upon activation some of the intact holoenzyme undergoes disassembly to form smaller oligomeric species. The variation between the different replicates observed here is expected, as the release of dimers is a non-equilibrium process, with different amounts of dimeric species present in the solution at the point of their capture on the glass. There are also oligomers of intermediate size, between dimer and dodecamer (as well as larger aggregates), present in the sample and these are not captured by the three-component mixture model. We estimated the deviation of the three-component distribution from the raw data (shown as residuals in *Figure 8B*), as a measure of the amount of intermediate species present, and observe an increase in the species between dimer and dodecamer upon activation.

Gel filtration experiments indicate that introduction of a mutation at the hub interface (F397A) leads to destabilization of the hub, with the release of dimers from the isolated hub (*Bhattacharyya et al., 2016*). We used single-molecule analysis to consider the effect of activation upon CaMKII-α stability when Phe 397 at the hub interface is mutated to alanine (F397A) (*Figure 9A*). The distribution at lower intensities for this sample has two clearly resolvable peaks that correspond to a pure dimer population with 30% dark GFP (*Figure 9B*). The unactivated sample of this mutant form has a much larger percentage of smaller species when compared to any of the wild-type samples (*Figure 9B*, *Figure 9—figure supplement 1*). Upon activation, the peak corresponding to intact holoenzymes is no longer observed (*Figure 9*, *Figure 9—figure supplement 1*), suggesting that the activation leads to complete destabilization of the CaMKII F397A variant holoenzyme.

## Conclusions

The unusual oligomeric organization of CaMKII, in which 12 or more kinase domains are arranged around a central hub, is coupled to a mechanism involving phosphorylation-mediated acquisition of constitutive activity that is also unique. When unphosphorylated, CaMKII is dependent on $Ca^{2+}$/CaM for kinase activity. Once phosphorylated at Thr 286, CaMKII subunits acquire calmodulin-independent activity (autonomy). Thr 286 phosphorylation must necessarily occur in trans, with one kinase subunit phosphorylating another. Efficient Thr 286 phosphorylation occurs only within a holoenzyme (*De Koninck and Schulman, 1998*), with the enzyme-kinase and substrate-kinase being tethered to the same hub at high local concentrations. This feature makes subunit exchange potentially important for spreading the activation of CaMKII, since it provides a means for activated subunits to enter into holoenzymes that have not yet been activated and thereby produce more activated subunits by trans-phosphorylation (*Stratton et al., 2014*).

The intriguing aspect of subunit exchange in CaMKII is that it is activation dependent. Most oligomeric proteins are capable of subunit exchange, provided that the dissociation rate constants are not too slow, and the subunits do not unfold when they are dissociated. Indeed, a CaMKII enzyme from choanoflagellates does not form stable oligomeric species, and it undergoes spontaneous subunit exchange (*Bhattacharyya et al., 2016*). In contrast, mammalian CaMKII-α is very stable as a dodecameric or tetradecameric oligomer and it demonstrates an insignificant degree of subunit exchange when not activated (*Stratton et al., 2014*; *Bhattacharyya et al., 2016*).

In this paper, we have shown, using electrospray ionization mass spectrometry, that the stable assembly formed by the CaMKII-α hub is destabilized by the addition of peptides derived from the regulatory segment of CaMKII-α. Addition of regulatory segment peptides, in both phosphorylated and unphosphorylated forms, at 10 μM or higher concentration to samples of the hub leads to rapid loss of integrity of the hub, and the release of monomers, dimers, tetramers, hexamers and octamers. Molecular dynamics simulations indicate that individual subunits undergo considerable displacements with respect to adjacent subunits. These transient fluctuations lead to the interfacial grooves between subunits opening up, allowing regulatory segments to be captured. The simulations indicate that although two regulatory segments can dock on a closed hub, the binding of additional segments could disrupt the integrity of the hub. These simulations are relatively short, and much longer simulations would be required to model the complete disassembly process. We have also shown, using a single-molecule TIRF assay, that activation of CaMKII-α expressed in mammalian cells results in destabilization of the holoenzymes.

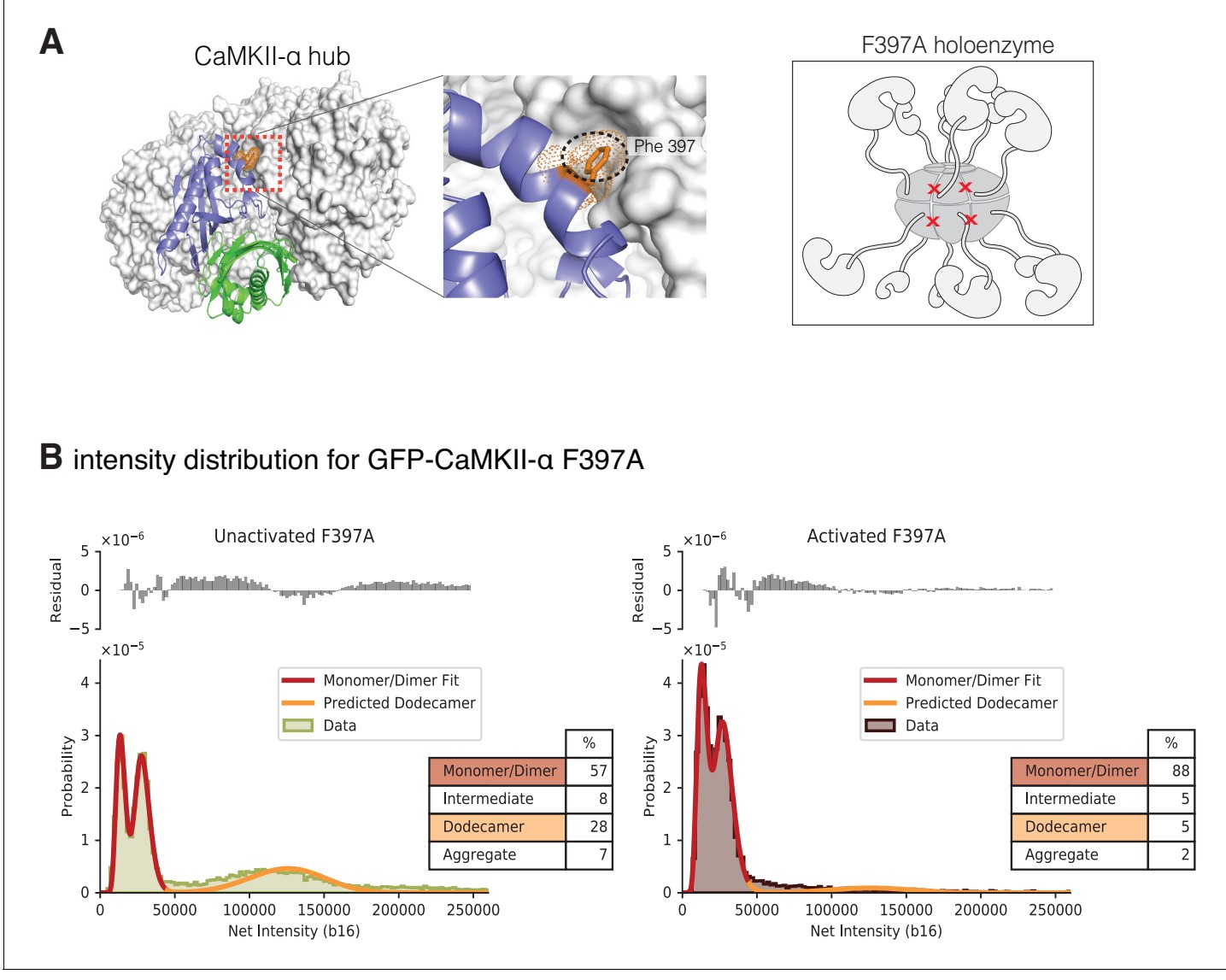

**Figure 9.** Intensity distribution of F397A mEGFP-CaMKII-α. (A) Crystal structure of human dodecameric CaMKII-α hub assembly (PDB ID – 5IG3) (*Bhattacharyya et al., 2016*) with the interface high-lighted. Replacement of Phe 397 at the hub interface with alanine weakens this interface. (B) Distribution of the intensities of unactivated (left) and activated (right) F397A mEGFP-CaMKII-α. At lower intensity, peaks corresponding to monomeric and dimeric species are observed. In the unactivated sample the peak at higher intensity, that corresponds to intact dodecameric holoenzyme is much smaller than in the wild-type sample. Upon activation this peak is no longer observed. As in the experiments with wild-type CaMKII-α, the intensity data were fit to a mixture comprising of monomers and dimers (red line) and the position of the peak corresponding to dodecamers was predicted (orange line). The residual plot shows the deviation from data with the fitted three-component distribution and shows the intermediate oligomeric species (between dimer and dodecamer). The percentage of the smaller, intermediate, dodecameric and aggregate species in the samples, as estimated from the fit, are shown.

The online version of this article includes the following source data and figure supplement(s) for figure 9:

**Source data 1.** Numerical data plotted in *Figure 9* and accompanying figure supplement.
**Figure supplement 1.** Intensity distribution of F397A mEGFP-CaMKII-α replicates.

We had suggested previously that there is a three-way competition for binding to the calmodulin-binding element between the kinase domain, Ca²⁺/CaM and the hub interface (*Bhattacharyya et al., 2016*). The calmodulin-binding element binds with nanomolar affinity to the kinase domain (*Colbran et al., 1988*), with picomolar affinity to Ca²⁺/CaM (*Tse et al., 2007*), and with micromolar affinity to the hub (*Bhattacharyya et al., 2016*). Upon activation, phosphorylation on Thr 286 prevents the calmodulin-binding element from binding to the kinase domains and

phosphorylation on Thr 305/Thr 306 prevents interaction with Ca$^{2+}$/CaM. Thus, if a CaMKII subunit undergoes phosphorylation at both Thr 286 and at Thr 305/Thr 306, then the regulatory segment might be released, and free to interact with the hub interface. The local concentration of the regulatory segments with respect to the hub is estimated to be in the millimolar range (*Bhattacharyya et al., 2016*). Given the observation that the addition of the regulatory segment peptides at micromolar concentrations in trans leads to hub disassembly, we conclude that the presentation of these elements in cis to the hub could lead to rapid destabilization of the hub upon activation.

Subunit exchange in CaMKII has been proposed originally as a mechanism whereby the activation states of CaMKII could be maintained indefinitely, contributing to the storage of memories (*Lisman, 1994*). It is difficult to reconcile such a mechanism with the action of phosphatases, which are expected to reset the activation sates of CaMKII (*Lee et al., 2009*; *Bhattacharyya et al., 2020b*). Instead, we speculate that the destabilization of CaMKII holoenzymes upon activation is a mechanism that potentiates the effects of Ca$^{2+}$ pulses under conditions when calmodulin is limiting, as is the case in neurons (*Persechini and Stemmer, 2002*; *Pepke et al., 2010*). Under these conditions, the release of dimers and other smaller oligomers can result in the spread of activated CaMKII species that can interact with and activate as yet unactivated holoenzymes. The precise mechanism by which this spread of activation occurs, and the role of subunit exchange, awaits further study.

# Materials and methods

## Key resources table

| Reagent type (species) or resource | Designation | Source or reference | Identifiers | Additional information |
|---|---|---|---|---|
| Gene (*Homo sapiens*) | CAMK2N1 | GenBank | HGNC:HGNC:1460 | |
| Strain, strain background (*Escherichia coli*) | BL21(DE3) | Sigma-Aldrich | CMC0014 | Chemically competent |
| Cell line (*Homo sapiens*) | HEK293T | UC Berkeley cell culture facility | | Authenticated using STR profiling and tested negative for mycoplasma |
| Recombinant DNA reagent | pEGFP-C1 (plasmid) | Clontech, Mountain View, CA | | Vector backbone for inserting the CaMKII genes |
| Recombinant DNA reagent | pSNAP$_f$ (plasmid) | New England Biolabs, MA | N9183S | Vector backbone |
| Transfected construct (*Homo sapiens*) | pEGFP-C1-Avi-mEGFP-CaMKII | This paper | | pEGFP-C1 vector was modified to have a biotinylation sequence followed by a linker at the N-terminus of mEGFP. Full-length human CaMKII-α is at the C-terminus of mEGFP. |
| Transfected construct (*Homo sapiens*) | pET21a-BirA | Addgene | # 20857 | |
| Transfected construct | pcDNA3.1 | Addgene | | |
| Recombinant DNA reagent | pET 28-6xHis-precission - CaMKII Hub | This paper | | pET-28 vector was modified to contain a PreScission Protease site between the N-terminal 6-histidine tag and the Human CaMKII-α hub coding sequence |
| Recombinant DNA reagent | pSNAPf- BirA | This paper | | BirA was cloned into the pSNAPf vector after removing the SNAP-tag. |

*Continued on next page*

*Continued*

| Reagent type (species) or resource | Designation | Source or reference | Identifiers | Additional information |
|---|---|---|---|---|
| Peptide, recombinant protein | Peptide A (LKKFNARRKLKGAILTTMLA) | Elim Biopharm | | |
| Peptide, recombinant protein | Peptide B (AILTTMLATRNFSGGKSGGNKK) | Elim Biopharm | | |
| Peptide, recombinant protein | Peptide C (LKEFNAERKLEGAILTTMLA) | Elim Biopharm | | |
| Peptide, recombinant protein | Peptide D (LKKFNARRKLKGAGSGSGSAC) | Elim Biopharm | | |
| Peptide, recombinant protein | Peptide G (RSTVASCMHRQETVDCLKKFNA) | Elim Biopharm | | |
| Peptide, recombinant protein | Peptide F (LKKFNARRKLKGAILGSMLAC) | Elim Biopharm | | |
| Peptide, recombinant protein | Peptide H, CaMKIINtide (KRPPKLGQIGRSKRVVIEDDRIDDVLK) | Elim Biopharm | | |
| Peptide, recombinant protein | Peptide A$^{phos}$ (LKKFNARRKLKGAILpTTMLA) | Elim Biopharm | | |
| Peptide, recombinant protein | Peptide B$^{phos}$ (AILpTTMLATRNFSGGKSGGNKK) | Elim Biopharm | | |
| Peptide, recombinant protein | Poly-L-lysine PEG (PLL:PEG) | SuSoS, Dübendorf, Switzerland | PLL(20)-g[3.5]- PEG(2) | Preparation of flow chambers |
| Peptide, recombinant protein | streptavidin | Sigma-Aldrich | S0677 | Functionalize the glass substrates for capturing biotinylated CaMKII |
| Peptide, recombinant protein | Calmodulin | Sigma-Aldrich | C4874 | Activation of CaMKII |
| Chemical compound, drug | PEG-Biotin | SuSoS, Dübendorf, Switzerland | PLL(20)-g[3.5]-PEG(2)/PEG(3.4)-biotin(50%) | Preparation of flow chambers |
| Chemical compound, drug | 1% protease inhibitor cocktail | Sigma | P8340 | Protease inhibitor cocktail for lysis buffer |
| Chemical compound, drug | 0.5% phosphatase inhibitor cocktail 2 and 3 | Sigma | P0044 and P5726 | Phosphatase inhibitor cocktails for lysis buffer |
| Software, algorithm | FIJI (ImageJ) | Open access software, see https://imagej.net/Fiji/Downloads | | Image processing |
| Software, algorithm | In-house python code | Open access, see, See *Figure 8* and *9* *Figure 8—source code 1* | | Image processing |
| Other | Sticky-Slide VI 0.4 | Ibidi | 80608 | Flow chambers |
| Other | Glass coverslips | Ibidi | 10812 | functionalized substrates |
| Software, algorithm | Mass spectrometry data acquisition | Software MassLynx 4.1 SCN 957 | | Data collected on a Waters SYNAPT G2Si Mass spectrometer. Version 4.1 SCN 957 or later required to process data |
| Other | 500 nm opening borosilicate emitters | | | Generated in the lab with a tip puller |

## Preparation of plasmids

For the mass spectrometry analyses, the human CaMKII-α hub domain (Uniprot_ID: Q9UQM7, residue 345–475) was cloned into a pET-28 vector (Novagen) that was modified to contain a PreScission Protease (Pharmacia) site between the N-terminal 6-histidine tag and the coding sequence, before being expressed in *E. coli*. For the single-molecule analyses, the pEGFP-C1 vector was modified to

contain a biotinylation sequence (Avitag, GLNDIFEAQKIEWHE) followed by a linker (GASGASGAS-GAS) at the N-terminus of mEGFP. Full-length human CaMKII-α (Uniprot_ID: Q9UQM7) was cloned into this vector backbone (Clontech), at the C-terminus of mEGFP, with a linker sequence (PreScission protease site: LEVLFQGP) separating the mEGFP tag from the coding sequence of CaMKII-α, before being overexpressed in mammalian cells (*Bhattacharyya et al., 2020b*). This construct was then used as a template to produce a CaMKII-α variant (CaMKII-α-F397A). pET21a-BirA, which carries the *E. coli* biotin protein ligase sequence, was a gift from Alice Ting (Addgene plasmid # 20857). BirA was cloned into the pSNAP$_f$ vector (New England Biolabs) after modifying the vector to remove the SNAP-tag. All constructs with insertions and deletions were made using standard protocols for Gibson assembly (New England Biolabs). All point mutants used were generated using standard Quikchange protocols (Agilent Technologies).

## Expression and purification of CaMKII-α hub

Human CaMKII-α hub domain was expressed in *E. coli* and purified as previously described (*Bhattacharyya et al., 2016*). Briefly, CaMKII-α hub protein expression was carried out in *E. coli* BL21 cells. Cells were induced by the addition of 1 mM isopropyl β-D-1-thiogalactopyranoside at an optical density of 0.7–0.8 and grown overnight at 18°C. Cell pellets were resuspended in Buffer A (25 mM Tris, pH 8.5, 150 mM potassium chloride, 1 mM DTT, 50 mM imidazole, and 10% glycerol) and lysed using a cell homogenizer. The cell lysate was filtered through a 1.1 micron glass fiber pre-filter (ThermoFisher). The filtered lysate was loaded onto a 5 mL Ni-NTA column and eluted with Buffer B (25 mM Tris, pH 8.5, 150 mM potassium chloride, 1 mM DTT, 0.5 M imidazole, and 10% glycerol). The eluate was desalted using a HiPrep 26/10 desalting column into Buffer A with 10 mM imidazole, and cleaved with PreScission protease (overnight at 4°C). The cleaved samples were loaded onto a Ni-NTA column, the flow through was collected, concentrated and purified further using a Superose six gel filtration column equilibrated in gel filtration buffer (25 mM Tris, pH 8.0, 150 mM KCl, 1.0 mM tris(2-carboxyethyl)phosphine [TCEP] and 10% glycerol). Fractions with pure protein were pooled, concentrated and stored at −80°C. All purification steps were carried out at 4°C, and all columns were purchased from GE Healthcare (Piscataway, NJ).

## Electrospray ionization mass spectrometry

Stock solutions of CAMKII-α hub were buffer-exchanged into 1 M ammonium acetate using a Bio-Spin column (Bio-Spin 6, Bio-Rad Laboratories, Inc, Hercules, CA). 10 mM stock solutions of different regulatory segment-derived unphosphorylated and phosphorylated peptides were prepared by dissolving the lyophilized powder in ddH$_2$O. All peptides used in this study were purchased from Elim Biopharm, Hayward, CA. The hub and peptide were mixed such that final concentrations in the reaction mix contained 120 μM CaMKII-α hub (subunit concentration) and 1 μM to 1 mM of the peptide, with a constant ionic strength of ~400 mM of ammonium acetate. This reaction mix was incubated for 5 min at room temperature before analysis by electrospray ionization mass spectrometry. In experiments of the hub incubated with phosphorylated peptides, the mass spectra obtained comprised predominantly of smaller oligomeric species and aggregates. Therefore, the last incubation step was eliminated, and the sample was introduced into the mass spectrometer as soon as the hub was mixed with the phosphorylated peptide. As a control experiment, 120 μM of CaMKII-α hub in 400 mM ammonium acetate was also incubated for five minutes at room temperature and analyzed using electrospray ionization mass spectrometry.

Mass spectra were acquired on a Waters SYNAPT G2-Si mass spectrometer (Waters, Milford, MA). Ions were formed from borosilicate emitters with a tip diameter of ~500 nm that were pulled from capillaries (1.0 mm o.d./0.78 mm i.d., Sutter Instruments, Novato, CA) using a Flaming/Brown micropipette puller (Model P-87, Sutter Instruments, Novato, CA). The submicron emitters were chosen to help reduce salt adducts on the protein ions and to reduce the chemical noise (*Susa et al., 2017*). Nanoelectrospray ionization was initiated by applying a voltage of ~1.0 kV to 1.2 kV onto a platinum wire (0.127 mm in diameter, Sigma, St. Louis, MO) that is inserted inside the emitter and in contact with sample solutions. The emitter tips were placed ~6 mm to 8 mm away from the entrance of the mass spectrometer.

To minimize collisional activation of the ions, a relatively low-sample cone voltage (50 V) and low-source temperature (80°C) were used. The voltages of collisional induced dissociation cells (both

Trap and Transfer cells) were set at 2 V, a value at which no gas-phase dissociation of the protein complex was observed. Mass spectral data were smoothed using the Savitsky-Golay smoothing algorithm, available via the Waters MassLynx software, with a smoothing window of 100 $m/z$ (mass-to-charge ratio). For all experiments, the mass spectral results were continuously recorded as long as protein signals were observed. Mass spectra of the hub incubated with peptides were generated by averaging the scans for specific time segments after the initiation of electrospray ionization. These times correspond to the start of the mass spectrometry data acquisition and do not include the initial incubation time.

## Analytical gel filtration chromatography

For analytical gel filtration chromatography, stock solution of CAMKII-α hub was diluted to 120 μM (subunit concentration) in 1 M ammonium acetate to maintain the same conditions with mass spectrometry. 10 mM stock solution of Peptide A was prepared by dissolving the lyophilized powder in ddH$_2$O. The hub and peptide were mixed in 1M ammonium acetate to achieve final concentrations of 120 μM CaMKII-α hub (subunit concentration) and 500 μM of the peptide. This reaction mix was incubated for 15 min at room temperature before loading onto the gel filtration column. Samples (hub alone or hub+peptide) were loaded onto a Superose 6 10/300 column (10/300 GL; GE Healthcare) equilibrated with 1M ammonium acetate at pH 7, at a flow rate of 0.5 ml/min (Prominence UFLC, Shimadzu).

## System preparation for molecular dynamics simulations

For the long-timescale simulations, we built a dodecameric hub assembly by first building a model of a vertical dimer of hub domains connected to the regulatory segments by unstructured linkers. We ran short simulations of this structure while holding the hub domains fixed and allowing the linkers and regulatory segments to sample various conformations. We used instantaneous structures from these short simulations to build the dodecameric hub assembly with linkers and regulatory segments, to ensure that the simulation started with the regulatory segments in a variety of conformations at varying distances from the hub assembly.

For the short simulations of a vertical dimer, one vertical dimer of hub domains (residues 345–472) was taken from the crystal structure of the human CaMKII-α dodecamer (PDB ID - 5IG3) (*Bhattacharyya et al., 2016*). Residues 336–344 at the N-terminus of the hub domain are missing in this structure and were modeled based on the structure of a tetradecameric CaMKII-α hub (PDB ID - 1HKX *Hoelz et al., 2003*). The C-terminus of the hub domain (resides 473–478) and the linker (residues 314–335) were built in irregular conformations using Coot (*Emsley and Cowtan, 2004*) and PyMOL (*Schrödinger, LLC, 2015*), via the SBGrid Consortium (*Morin et al., 2013*). The N-terminal (residues 281–300) and C-terminal regions (residues 308–313) of the regulatory segment were modeled based on the structure of a CaMKII-calmodulin complex (PDB ID - 2WEL) (*Rellos et al., 2010*). Residues 301–307, which comprise the β-clip segment of the calmodulin-binding element, form a helix when bound calmodulin (*Rellos et al., 2010*). In the absence of calmodulin, this region is expected to unwind from a helical architecture, and so the starting structure for this region was modeled based on the structure of an autoinhibited CaMKII, where this portion of the helix is unwound (PDB ID - 2VN9) (*Rellos et al., 2010*). The N-termini of the individual subunits were capped with acetyl groups, because they represent truncations of the actual structure, whereas the C-termini were left uncapped.

The protein was solvated, and ions were added such that the ionic strength was 0.15 M, using VMD (*Humphrey et al., 1996*). The energy of this system was minimized for 1000 steps, first while holding the protein atoms fixed, then for 1000 steps while allowing all the atoms to move. It was then equilibrated by molecular dynamics at constant pressure (1 atm) and constant temperature (300 K) for 2 ns, then equilibrated at constant volume and constant temperature (300 K), to ensure that the pressure is maintained stably, for 1 ns. We generated two short trajectories (each ~60 ns) at constant pressure (1 atm) and constant temperature (300 K), while holding hub domains (residues 336–478) fixed, and allowing the linkers and regulatory segments to explore various conformations.

Instantaneous structures from these simulations, with the linker and regulatory segment in arbitrary conformations, were then used to build a dodecameric assembly comprising of the hub, linker and regulatory segments, by aligning a pair of vertical hub domain dimers onto the vertical hub

domain dimers of the human dodecameric hub assembly (PDB ID – 5IG3) (*Bhattacharyya et al., 2016*). This system was then solvated and ions were added so that the final ionic strength was 0.15 M using VMD (*Humphrey et al., 1996*). The final system had a cubic cell with edge dimensions of 193 Å.

A second system comprising of the dodecamer hub assembly only, without the linker and regulatory segments was built from the system described above, by removing residues 281–335 from each subunit. This system was solvated with TIP3P water and ions were added so that the final ionic strength was 0.15 M using VMD (*Humphrey et al., 1996*). The final system had a cubic cell with edge dimensions of 193 Å.

The energies of both systems were minimized first while holding the protein fixed, then while allowing all the atoms to move. The two systems were then equilibrated at constant pressure (1 atm) and temperature (300 K) for 5 ns. The NAMD package was used to run the minimization and equilibration simulations (*Phillips et al., 2005*) with the CHARMM36 force field (*Best et al., 2012*). The velocity Verlet algorithm was used to calculate the trajectories of the atoms. A time step of 2 fs was used. Particle Mesh Ewald was used to calculate long-range electrostatic interactions, with a maximum space of 1 Å between grid points (*Darden et al., 1993*). Long-range electrostatics were updated at every time step. Van der Waal's interactions were truncated at 12 Å. Hydrogen atoms bonded to the heavy atoms were constrained using the ShakeH algorithm (*Ryckaert et al., 1977*). Temperature was controlled with Langevin dynamics with a damping coefficient of 4/ps, applied only to non-hydrogen atoms. Pressure was controlled by the Nose-Hoover method with the Langevin piston, with an oscillation period of 200 fs and a damping time scale of 50 fs (*Martyna et al., 1994*; *Feller et al., 1995*).

## Anton simulation protocols

Production runs for both systems were carried out on Anton2 (*Shaw et al., 2014*). The restart files from the end of the equilibration stage were converted to the Desmond format (*Bowers et al., 2006*) using the VMD (*Humphrey et al., 1996*). The CHARMM param36 force field (*Best et al., 2012*) was used through the Viparr utility of Anton2 (*Shaw et al., 2014*). A time step of 2 fs was used. The multigrator integration method (*Lippert et al., 2013*) was used to generate the trajectories while using the Nose-Hoover thermostat to maintain temperature at 300 K and the MTK barostat to maintain pressure maintain pressure at one atm (*Lippert et al., 2013*). Coordinates were saved every 0.24 ns. The coordinate trajectories were converted to the NAMD DCD format using VMD (*Humphrey et al., 1996*).

The angle between the αA helices (residues 341–363) of adjacent subunits were calculated using the CHARMM package (*Brooks et al., 2009*). The r. m. s. deviation for each hub domain over the course of the simulation of the dodecamer with the regulatory segment and linkers present was calculated with the AmberTools18 package (*Case et al., 2018*), using residues 340–470 of the hub domain while fitting to the structure of the hub at the start of the simulation.

## Normal mode analysis

The hub assembly (residues 340:470) from the structure of the dodecameric autoinhibited CaMKII-α (PDB ID – 3SOA) was used for normal mode analysis. The *elNémo* server was used for the analysis with a minimum and maximum perturbation of 400, a step-size of 16 and a residue-block size of 8 (Suhre and Y.-H. *Suhre and Sanejouand, 2004a*; *Suhre and Sanejouand, 2004b*). The displacement vectors were displayed using the modevectors script in PyMOL (*Schrödinger, LLC, 2015*).

## Tissue culture and DNA transfection

HEK 293 T cells (UC Berkeley cell culture facility) were grown in DMEM (Dulbecco's Modified Eagle Medium + GlutaMaX, ThermoFisher) supplemented with 10% fetal bovine serum (FBS), 1X antibiotic–antimycotic reagent (ThermoFisher) and 20 mM HEPES buffer and maintained at 37°C under 5% $CO_2$. CaMKII-α variants were transiently transfected using the standard calcium phosphate protocol. Briefly, calcium phosphate/DNA coprecipitate was prepared by combining CaMKII-α plasmids with 5 μg of empty pcDNA3.1 vector and 1 μg of BirA plasmid. The mixture was then diluted with 10X ddH$_2$O, and $CaCl_2$ was added such that the final $CaCl_2$ concentration is 250 mM. This mixture was incubated for 15 min. One volume of this 2X calcium phosphate/DNA solution was added to an

equal volume of 2X HEPES-buffered saline (HBS) (50 mM HEPES, 280 mM NaCl, 1.5 mM Na$_2$HPO$_4$, pH 7.1) and the solution was mixed thoroughly by reverse pipetting. This mixture was then added to HEK 293 T cells and the cells were allowed to express the protein for 18–20 hr, before the protein was harvested.

## Preparation of flow cells for single-molecule microscopy

All single-molecule experiments were performed in flow chambers (sticky-Slide VI 0.4, Ibidi) that were assembled with functionalized glass substrates. The glass substrates (coverslips, Ibidi) were first cleaned using 2% Hellmanex III solution (Hellma Analytics) for 30 min, followed by a 30 min sonication in 1:1 mixture (vol/vol) of isopropanol:water. The glass substrates were then dried with nitrogen and cleaned for another 5 min in a plasma cleaner (Harrick Plasma PDC-32 G). These cleaned glass substrates were used to assemble the flow chambers immediately after plasma cleaning. After assembly, the glass substrates were treated with a mixture of Poly-L-lysine PEG and PEG-Biotin (1000:1, both at 1 mg/mL) for 30 min (SuSoS). The glass substrates were then washed with 2 mL of phosphate-buffered saline (PBS). Streptavidin was added to these glass substrates at a final concentration of 0.1 mg/mL and incubated for 30 min. Following incubation, excess streptavidin was washed away using 2 mL of PBS and these assembled flow chambers were used for all our single-molecule experiments.

## Cell lysis and pulldown of biotinylated CaMKII-α in flow chambers

The co-expression of the *E. coli* biotin ligase, BirA, with the CaMKII-α variants bearing an N-terminal Avitag, results in the biotinylation of the Avitag in HEK 293 T cells. After harvesting, the cells were lysed in a lysis buffer (25 mM Tris at pH 8, 150 mM KCl, 1.5 mM TCEP-HCl, 1% protease inhibitor cocktail (P8340, Sigma)), 0.5% phosphatase inhibitor cocktails 2 (P0044, Sigma) and 3 (P5726, Sigma), 50 mM NaF, 15 µg/ml benzamidine, 0.1 mM phenylmethanesulfonyl fluoride and 1% NP-40 (ThermoFisher).

The cell lysate was diluted in the gel filtration buffer (same recipe as above, but without glycerol). mEGFP-CaMKII-α in the diluted cell lysate was activated by incubating the lysate with an activation buffer containing 5 µM CaM, 100 µM CaCl$_2$, 10 mM ATP and 20 mM MgCl$_2$ (final concentrations in the reaction) for 60 min at room temperature. For the unactivated sample, the 60-min incubation was carried out in gel filtration buffer without the addition of any components of the activation buffer. For both the activated and unactivated samples, 100 µL of the diluted cell lysate was added to a well in the flow chamber. After incubation for 1 min, the diluted cell lysate was washed out with 1 mL of PBS. During this incubation, the biotinylated mEGFP-CaMKII-α variants were immobilized on the surface of the functionalized glass substrates, via the streptavidin-biotin interaction.

## Single-molecule total internal reflection fluorescence (TIRF) microscopy

Single-particle total internal reflection fluorescence (TIRF) images were acquired on a Nikon Eclipse Ti-inverted microscope equipped with a Nikon 100 × 1.49 numerical aperture oil-immersion TIRF objective, a TIRF illuminator, a Perfect Focus system, and a motorized stage. Images were recorded using an Andor iXon electron-multiplying CCD camera. The sample was illuminated using the LU-N4 laser unit (Nikon) with solid-state lasers for the channels emitting at wavelengths of 488 nm, 561 nm and 640 nm. Lasers were controlled using a built-in acousto-optic tunable filter. A 405/488/561/638 nm Quad TIRF filter set (Chroma Technology Corp.) was used along with supplementary emission filters of 525/50 m, 600/50 m, 700/75 m for the 488 nm, 561 nm and 640 nm channels, respectively. Image acquisition was performed with the automated change of illumination and filter sets, at 75 different positions from an initial reference frame, so as to capture multiple non-overlapping images, using the Nikon NIS-Elements software. mEGFP- CaMKII-α was imaged by illuminating the 488 nm laser set to 5.2 mW. Images for all the mEGFP-CaMKII-α variants were acquired using an exposure time of 80 ms, while keeping the laser power unchanged over all conditions.

To correct for the baseline clamp and dark current, a series of dark images were collected under the same exposure time of 80 ms as the experimental sample. Uneven illumination and fringe interference effects in the intensity distribution (*Mattheyses et al., 2010*) were corrected for by measuring the field illumination with a solubilized fluorescein sample. The dark image was subtracted from

both the sample data and field illumination control, then the sample data were divided by a normalized mean field control to acquire appropriate intensities.

For photobleaching experiments, the GFP-fluorescence signal was recorded in a stream acquisition mode, with an exposure time of 80 ms, under the same conditions as described above. As a control, the intensities and photobleaching traces of mEGFP-Hck, a monomeric protein kinase, were also collected under the same conditions.

## Analyses of single-molecule TIRF data

Individual single particles of mEGFP-CaMKII-$\alpha$ were detected and localized using the single particle tracking plugin TrackMate in ImageJ (*Jaqaman et al., 2008*). The particles were localized with the Laplacian of Gaussian detector with an initial diameter set to 6 pixels. A threshold value of 300 was used to exclude noisy, low-intensity spots. To eliminate the effects of variation in resolution at the edges of the field due to heterogenous TIRF illumination, only particles within a central area of 400 $\times$ 400 pixel$^2$ were included in the calculation. The intensity distributions for single particles of mEGFP-CaMKII-$\alpha$ (488 nm) were determined using custom in-house software written in Python. The total intensity values acquired from TrackMate for the 488-channel were adjusted by subtracting the local median background intensity for each spot (scaled by the spot area) to produce an intensity histogram for all the mEGFP-CaMKII-$\alpha$ spots.

For the analysis of the GFP step-photobleaching data, the photobleaching traces for each spot were built by plotting the maximum intensities of mEGFP-CaMKII-$\alpha$ as a function of time with MATLAB. The number of single photobleaching events were counted manually by inspecting the photobleaching trace for every spot. Single-step and two-step bleaching traces were clearly identified, but multistep photobleaching traces exhibited relatively unclear, distinct bleaching events. We therefore categorized the bleaching traces into single-step, two-step, and multi-step photobleaching. A similar analysis was used for mEGFP-Hck, which exhibited only single step traces.

## Determination the composition of oligomeric species in the intensity distributions

The intensity distribution of monomeric GFP species is lognormal when detected on an EMCCD camera, and is given by

$$p_M(x; \mu, \sigma) = \frac{1}{x\sigma\sqrt{2\pi}} e^{\frac{-(\ln x - \mu)^2}{2\sigma^2}}$$

where $x$ is the integrated intensity of a spot, $\mu$ is the mean of the underlying normal variable and $\sigma$ is its standard deviation. Given a monomer distribution, the intensity distribution for a species with N subunits can be determined by convolving the intensity distribution of the monomer with itself N times (*Mutch et al., 2007*). An oligomer may have some dark GFP leading to a different intensity distribution than expected, and the dark fraction ($\nu$) can be used to predict the extent to which this occurs. This was done by sampling $p_M$ finely, converting it to a probability mass function (pmf) and setting

$$p_M^{pmf}(x = 0) = \nu$$

Subsequent convolutions of this modified probability mass function carry the effect of the dark GFP. The probability mass function was then reconverted back to a density function.

We estimated the dark fraction ($\nu$) by auto-convolving the monomeric distribution 12 times and determining value of $\nu$ that yields the best alignment of the predicted dodecameric distribution with the high-intensity peak in the experimental data. These gave values of $\nu$ ~0.3 (*Table 1*). While this method of estimating $\nu$ will center the predicted peak with the data, it has no bearing on its width, which is governed by the underlying lognormal function. Based on these results, a dark fraction of 0.3 was used for all samples.

The following procedure was used to create the final mixture model in *Figures 8* and *9*. First the low-intensity peak was fit to a mixture of monomeric and dimeric species with a fixed dark fraction ($\nu$) of 0.3, which yielded parameters $\sigma$ and $\mu$ for the underlying monomeric lognormal distribution. This log-normal distribution was convolved with itself 12 times, also with a fixed dark rate of 0.3, to

gives three final distributions: monomer, dimer, and dodecamer. The observed data were then fit to a mixture of these three populations.

## Acknowledgements

We thank Howard Schulman and Christine Gee for helpful discussions. ZX and ERW are supported by the National Science Foundation Division of Chemistry under grant number CHE-1609866. The authors also thank CalSOLV for funding. MB thanks NIGMS (K99 GM 126145) for funding. Preliminary simulations were run on Comet, at the San Diego Supercomputing Center, via Extreme Science and Engineering Discovery Environment (XSEDE), which is supported by National Science Foundation grant number ACI-1548562. Long-timescale simulations were carried out on Anton2. Anton2 computer time was provided by the Pittsburgh Supercomputing Center (PSC) through Grant R01GM116961 from the National Institutes of Health. The Anton2 machine at PSC was generously made available by D.E. Shaw Research.

## Additional information

### Competing interests

John Kuriyan: Senior editor, *eLife*. The other authors declare that no competing interests exist.

### Funding

| Funder | Grant reference number | Author |
|---|---|---|
| National Institute of General Medical Sciences | K99 GM 126145 | Moitrayee Bhattacharyya |
| National Science Foundation | CHE-1609866 | Zijie Xia<br>Evan R Williams |
| Howard Hughes Medical Institute | | John Kuriyan |
| calsolv | | Zijie Xia<br>Evan R Williams |
| National Institute of General Medical Sciences | K99 GM 126145 | Moitrayee Bhattacharyya |

The funders had no role in study design, data collection and interpretation, or the decision to submit the work for publication.

### Author contributions

Deepti Karandur, Zijie Xia, Conceptualization, Resources, Data curation, Software, Formal analysis, Validation, Investigation, Visualization, Methodology, Writing - original draft, Writing - review and editing; Moitrayee Bhattacharyya, Conceptualization, Resources, Data curation, Software, Formal analysis, Funding acquisition, Validation, Investigation, Visualization, Methodology, Writing - original draft, Writing - review and editing; Young Kwang Lee, Serena Muratcioglu, Data curation, Formal analysis, Investigation, Methodology, Writing - original draft, Writing - review and editing; Darren McAffee, Resources, Data curation, Software, Formal analysis, Validation, Visualization, Methodology, Writing - original draft, Writing - review and editing; Ethan D McSpadden, Baiyu Qiu, Data curation, Formal analysis, Investigation; Jay T Groves, Evan R Williams, John Kuriyan, Conceptualization, Resources, Data curation, Software, Formal analysis, Supervision, Funding acquisition, Validation, Investigation, Visualization, Methodology, Writing - original draft, Project administration, Writing - review and editing

### Author ORCIDs

Deepti Karandur ⬛ https://orcid.org/0000-0002-6949-6337
Moitrayee Bhattacharyya ⬛ https://orcid.org/0000-0002-2168-1541

Young Kwang Lee https://orcid.org/0000-0003-0056-6357
John Kuriyan https://orcid.org/0000-0002-4414-5477

**Decision letter and Author response**
Decision letter https://doi.org/10.7554/eLife.57784.sa1
Author response https://doi.org/10.7554/eLife.57784.sa2

## Additional files

### Supplementary files
• Transparent reporting form

### Data availability

Molecular dynamics simulation trajectories are available at Pittsburg Supercomputing Center's data storage facility and are accessible at the following link: https://psc.edu/anton-project-summaries?id=3071&pid=35. Mass spectrometry data (Figure 2-4) is available via the MassIVE database under identifier MSV000086103.

The following datasets were generated:

| Author(s) | Year | Dataset title | Dataset URL | Database and Identifier |
| --- | --- | --- | --- | --- |
| Kuriyan J | 2020 | Breakage of the Oligomeric CaMKII Hub by the Regulatory Segment of the Kinase | https://psc.edu/anton-project-summaries?id=3071&pid=35 | Pittsburg Supercomputing Center, 3071 |
| Kuriyan J | 2020 | Breakage of the Oligomeric CaMKII Hub by the Regulatory Segment of the Kinase | https://massive.ucsd.edu/ProteoSAFe/dataset.jsp?accession=MSV000086103 | MassIVE, MSV000086103 |

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
