## [Decision Letter]

**Acceptance summary:**

The calcium/calmodulin-dependent kinase II (CamKII) plays key roles in the brain and heart. This paper uses an elegant combination of mass spectrometry, computational analysis, and cellular experiments to extend our understanding of the dynamics of the oligomeric structure of CamKII. Using synthetic peptides derived from the CaMKII regulatory segment, the authors show that the highly organized Hub structure of this kinase can be destabilized providing a molecular mechanism of controlling its cellular function.

**Decision letter after peer review:**

Thank you for submitting your article "Breakage of the Oligomeric CaMKII Hub by the Regulatory Segment of the Kinase" for consideration by *eLife*. Your article has been reviewed by three peer reviewers, and the evaluation has been overseen by Leslie Griffith as Reviewing Editor and Philip Cole as the Senior Editor. The following individuals involved in review of your submission have agreed to reveal their identity: Richard Bayliss (University of Leeds) (Reviewer #2); Dorothee Kern (Reviewer #3).

The reviewers have discussed the reviews with one another and the Reviewing Editor has drafted this decision to help you prepare a revised submission.

As the editors have judged that your manuscript is of interest, but as described below request an additional experiment before it is published, we would like to draw your attention to changes in our revision policy that we have made in response to COVID-19 (https://elifesciences.org/articles/57162). First, because many researchers have temporarily lost access to the labs, we will give authors as much time as they need to submit revised manuscripts. We are also offering, if you choose, to post the manuscript to bioRxiv (if it is not already there) along with this decision letter and a formal designation that the manuscript is "in revision at *eLife*". Please let us know if you would like to pursue this option. (If your work is more suitable for medRxiv, you will need to post the preprint yourself, as the mechanisms for us to do so are still in development.)

Summary:

This paper builds on studies from the Kuriyan lab published in *eLife* that have suggested that CaMKII, which exists as a 12-mer or 14-mer complex, can exchange subunits upon activation by Ca/CaM. Here they show short peptides derived from the CaMKII regulatory segment can bind and destabilize the 12-mer using native ESI mass spec. Molecular dynamics simulations were also performed to model a mechanism for dissociation. Lastly, the authors show data from a single molecule assay that suggest that activation can cause dissociation.

Essential revisions:

The reviewers found that this was both an interesting and potentially important follow-up to the lab's previous *eLife* paper(s). They do feel, however, that the manuscript could be improved by additional data (essential revision 1 below) and by additional prose/analysis/figures to flesh out a couple other areas (essential revisions 2-4) below.

1) The authors' previous studies indicated that phosphorylation of Thr305 and/or Thr306 in the regulatory domain potentiates subunit exchange. Indeed, the authors conclude that phosphorylation at T286 and T305/6 disrupt the linker's interactions with kinase domain and Ca/CaM, respectively, so that it can then bind to hub. However, none of the peptides (A, B, C) used in the current work were phosphorylated. Their prior paper indicated that peptide phosphorylation does not directly affect binding to the hub, but it should be very informative to compare the effects of Thr305- and/or Thr306-phosphorylation of peptide A to the current data set for subunit disassembly using the peptide/MS method.

2) There is not enough detail in the figures to make a clear comparison of the distortions in the hub observed in the presence and absence of peptide. The reviewers were therefore not completely convinced about the relevance of these distortions to the mechanism of peptide-induced dissociation of the hub. The analysis of subunit-subunit changes is presented simply as angular changes, whereas the docking interface is shown in atomic detail. Would it not make sense to also show the interface interactions, and to present an analysis of the molecular contacts at these interfaces?

3) In the second part of the manuscript, binding of peptide to the hub is investigated via docking/long MD simulations on ANTON. Two peptides are found to bind in two interfaces between subunits leading to distortions of the hub rom its normal circular shape to an oval shape. However, the hub does not actually separate at any of the interfaces. The authors suggest that the docking of additional segments to the hub are needed for hub disassembly. Would not the binding of just these two peptides as shown in Figure 5 with full disruption of the hub interactions in these two interfaces result in hexamers that are seen in the MS data? Why would docking of additional segment be needed? Looking at the rough time-scale of disassembly from the MS data (in the minute regime), could the more likely explanation for the lack of disassembly be that the process is too slow relative to the simulation times? A discussion of the timecourse of the dissociation, as observed by ESI-MS, and how this relates to the timescale of the simulation should also be included.

4) The data purportedly demonstrating that activation of mammalian cell-expressed CaMKII holoenzymes are not completely convincing: there is no statistical analysis of the replicates and the changes in the area of the peaks appear rather minor in two of the three replicates. The effects, if any, on WT holoenzymes 60 min after activation appear to be very modest compared to the essentially complete disruption of the hubs that was observed within a few minutes of peptide addition. This would seem to be important to ascertain whether this apparent "holoenzyme dissociation" results from the specific mechanism defined in the prior studies, or from a more general instability of CaMKII over this extended incubation. Indeed, the surprisingly significant levels of monomer/dimer in the inactivated samples may result from an alternative more general instability, complicating interpretation. The authors need to acknowledge and comment on these issues and provide adequate data treatment.

---

## [Author Response]

Essential revisions:The reviewers found that this was both an interesting and potentially important follow-up to the lab's previous eLife paper(s). They do feel, however, that the manuscript could be improved by additional data (essential revision 1 below) and by additional prose/analysis/figures to flesh out a couple other areas (essential revisions 2-4) below.1) The authors' previous studies indicated that phosphorylation of Thr305 and/or Thr306 in the regulatory domain potentiates subunit exchange. Indeed, the authors conclude that phosphorylation at T286 and T305/6 disrupt the linker's interactions with kinase domain and Ca/CaM, respectively, so that it can then bind to hub. However, none of the peptides (A, B, C) used in the current work were phosphorylated. Their prior paper indicated that peptide phosphorylation does not directly affect binding to the hub, but it should be very informative to compare the effects of Thr305- and/or Thr306-phosphorylation of peptide A to the current data set for subunit disassembly using the peptide/MS method.

We thank the reviewers for this suggestion. We now include results from experiments that explore the effects of phosphorylated Peptide A and Peptide B on hub stability, as measured by electrospray mass spectrometry (Figure 4). We observed a more rapid decrease in hub stability upon addition of the phosphorylated Peptide A (Peptide A^phos^) at different concentrations, when compared to unphosphorylated Peptide A. A similar effect was observed when phosphorylated Peptide B (Peptide B^phos^) was mixed with the hub. In order to capture by mass spectrometry the more rapid destabilization upon the addition of Peptide A^phos^ or Peptide B^phos^, we had to eliminate the 5-minute incubation prior to introducing the sample into the mass spectrometer that was done in the original experiments with Peptide A and Peptide B. That is, mass spectral acquisition was started as soon as the hub was mixed with the peptide. As the reviewer points out, our earlier publication had noted that there is no apparent difference in affinity between phosphorylated and unphosphorylated peptides. Those earlier experiments did not consider the effects of the peptide on hub stability. That the phosphorylated peptides are more effective at breaking the hub was therefore not anticipated. A more thorough analysis of this effect awaits future study.

2) There is not enough detail in the figures to make a clear comparison of the distortions in the hub observed in the presence and absence of peptide. The reviewers were therefore not completely convinced about the relevance of these distortions to the mechanism of peptide-induced dissociation of the hub. The analysis of subunit-subunit changes is presented simply as angular changes, whereas the docking interface is shown in atomic detail. Would it not make sense to also show the interface interactions, and to present an analysis of the molecular contacts at these interfaces?

We thank the reviewers for raising this point and have included figures of the distortions of the hub at an interface that “opens” and at which docking occurs, as well as one of the interfaces that “closes” (Figure 6—figure supplement 2). We have also included a figure (Figure 6—figure supplement 3), where the regulatory segment that docked at one of the open interfaces was modeled onto a closed interface, showing how steric clashes would prevent this docking unless the interface undergoes further distortion, which might lead to destabilization and disassembly of the hub. Hydrogen bonds across interfaces are preserved in the simulation, irrespective of whether the interface opens or closes, and we have included a discussion and figures showing this (Figure 6—figure supplement 2). While it is likely, based on analyses in Bhattacharyya et al., 2016, that these interactions will break when the hub undergoes disassembly, this has not occurred on the timescale of this simulation.

3) In the second part of the manuscript, binding of peptide to the hub is investigated via docking/long MD simulations on ANTON. Two peptides are found to bind in two interfaces between subunits leading to distortions of the hub rom its normal circular shape to an oval shape. However, the hub does not actually separate at any of the interfaces. The authors suggest that the docking of additional segments to the hub are needed for hub disassembly. Would not the binding of just these two peptides as shown in Figure 5 with full disruption of the hub interactions in these two interfaces result in hexamers that are seen in the MS data? Why would docking of additional segment be needed? Looking at the rough time-scale of disassembly from the MS data (in the minute regime), could the more likely explanation for the lack of disassembly be that the process is too slow relative to the simulation times? A discussion of the timecourse of the dissociation, as observed by ESI-MS, and how this relates to the timescale of the simulation should also be included.

We agree with the reviewers that the simulation is not long enough to capture the complete course of disassembly. Binding by two regulatory segments might be sufficient to induce disassembly and we have included this in the Discussion. We have also included a figure of one of the “closed” interfaces with a regulatory segment modeled into the docked position to show the steric clashes that might prevent further docking (Figure 6—figure supplement 3) without further distortion and destabilization of the hub.

4) The data purportedly demonstrating that activation of mammalian cell-expressed CaMKII holoenzymes are not completely convincing: there is no statistical analysis of the replicates and the changes in the area of the peaks appear rather minor in two of the three replicates. The effects, if any, on WT holoenzymes 60 min after activation appear to be very modest compared to the essentially complete disruption of the hubs that was observed within a few minutes of peptide addition. This would seem to be important to ascertain whether this apparent "holoenzyme dissociation" results from the specific mechanism defined in the prior studies, or from a more general instability of CaMKII over this extended incubation. Indeed, the surprisingly significant levels of monomer/dimer in the inactivated samples may result from an alternative more general instability, complicating interpretation. The authors need to acknowledge and comment on these issues and provide adequate data treatment.

Due to the nature of these experiments with cell lysates, conventional statistical measures, such as the average and standard error, are really not applicable. Variations from run to run are dominated by systematic error sources (e.g. transfection rates, reaction stoppage conditions, etc.) that affect the degree to which the reaction progresses. To address these variations, we present data from multiple independent replicates (3 replicates) for each experiment. The consistent theme is that activation always increases the proportion of monomers, dimers and lower order oligomers. Figures 8 and 9 have been reformatted to emphasize this dominant feature of the data.

The effect of activation on CaMKII holoenzyme destabilization is actually not that small. If we look strictly at the increase of dimers, we see that it increases ~1.3-2 fold in each replicate, which is significant. This can come from a relatively small fraction of dodecamer dissociation, and the original figures did not readily convey this fact. We have now updated the Figures 8 and 9 to more clearly indicate the fold-changes in dimer fraction.

The extent of peptide-induced destabilization is different between mass spectrometry and single-molecule TIRF as these are measuring two different samples and under very different experimental conditions. In mass spectrometry, we are studying the hub (without the rest of the protein) after addition of the calmodulin-binding element (CBE) peptide in *trans*. In contrast, intact CaMKII holoenzyme is studied in the single-molecule experiments, where the CBE peptide segment is presented in *cis*. Also, the cell lysate experiments are done in the presence of saturating Ca^2+^/calmodulin, which binds to the regulatory segment (in a normal activation process *in situ* the calcium levels would drop after stimulation). In addition, the holoenzyme activation may also lead to some subunit exchange, where the released smaller oligomeric species may re-equilibrate with the holoenzyme.

The reviewer’s comments led us to appreciate the fact that the different experiments we have carried out represent different approximations of the physiologically relevant conditions, which are themselves not well understood. As noted above, the cell-based experiments are done in the presence of saturating Ca^2+^/calmodulin, which would normally subside to low levels after a calcium pulse. In these experiments, therefore, Ca^2+^/calmodulin protects the regulatory segment from phosphorylation on Thr 305/Thr 306, and also from interaction with the hub. In the mass spectrometry experiments it is surprising that we see so much breakage, given that the peptide is added in *trans*. Indeed, the incubation of the isolated hub with the CBE peptides at different concentrations does not lead to the detection of smaller oligomeric species by classical analytical gel filtration. In the gel filtration experiment, the peptide is separated from the hub as soon as the sample enters the column. It is important to note that classical biochemical experiments, such as gel filtration, may not detect destabilization, as it may be a subtle effect under those experimental conditions. In contrast, in the mass spectrometry experiment, the local concentration of peptides and hub may be very high in the electrospray droplets, and they may also be subjected to local heating. Note that such heating, if it occurs, does not lead to any detectable destabilization of the hub in the absence of the peptide. We have included this discussion in the revised manuscript and presented the gel filtration data as a supplementary figure (Figure 4—figure supplement 1). We believe this will inform future experiments trying to characterize subtle destabilization effects in CaMKII or other systems.